# Scutoids are a geometrical solution to three-dimensional packing of epithelia

Pedro Gómez-Gálvez [1], Pablo Vicente-Munuera [1], Antonio Tagua[1], Cristina Forja[1], Ana M. Castro[1], Marta Letrán[1], Andrea Valencia-Expósito [2], Clara Grima [3], Marina Bermúdez-Gallardo[1], Óscar Serrano-Pérez-Higueras[1], Florencia Cavodeassi[4,5], Sol Sotillos [2], María D. Martín-Bermudo [2], Alberto Márquez [3], Javier Buceta[6,7] & Luis M. Escudero [1]

As animals develop, tissue bending contributes to shape the organs into complex three-dimensional structures. However, the architecture and packing of curved epithelia remains largely unknown. Here we show by means of mathematical modelling that cells in bent epithelia can undergo intercalations along the apico-basal axis. This phenomenon forces cells to have different neighbours in their basal and apical surfaces. As a consequence, epithelial cells adopt a novel shape that we term "scutoid". The detailed analysis of diverse tissues confirms that generation of apico-basal intercalations between cells is a common feature during morphogenesis. Using biophysical arguments, we propose that scutoids make possible the minimization of the tissue energy and stabilize three-dimensional packing. Hence, we conclude that scutoids are one of nature's solutions to achieve epithelial bending. Our findings pave the way to understand the three-dimensional organization of epithelial organs.

[1] Departamento de Biología Celular, Universidad de Sevilla and Instituto de Biomedicina de Sevilla (IBiS), Hospital Universitario Virgen del Rocío/CSIC/Universidad de Sevilla, 41013 Seville, Spain. [2] CABD, CSIC/JA/UPO, Campus Universidad Pablo de Olavide, 41013 Seville, Spain. [3] Departamento de Matemática Aplicada I, Universidad de Sevilla, 41012 Seville, Spain. [4] Centro de Biología Molecular Severo Ochoa and CIBER de Enfermedades Raras. C/ Nicolás Cabrera 1, 28049 Madrid, Spain. [5] St. George's, University of London, Cranmer Terrace, SW17 0RE London, UK. [6] Bioengineering Department, Lehigh University, Bethlehem, PA 18018, USA. [7] Chemical and Biomolecular Engineering Department, Lehigh University, Bethlehem, PA 18018, USA. These authors contributed equally: Pedro Gómez-Gálvez, Pablo Vicente-Munuera. Correspondence and requests for materials should be addressed to J.B. (email: jbuceta@lehigh.edu) or to L.M.E. (email: lmescudero-ibis@us.es)

Epithelial cells are the building blocks of metazoa. These bricks display columnar, cubic, or squamous shapes and organize in simple or multilayer arrangements. Faithful execution of the body plan during morphogenesis requires a complex reshaping of epithelial tissues to achieve organ development. In this context, the transition from planar epithelial sheets to cylindrical, ellipsoidal, or spherical forms, involves fundamental reorganization of the cells along their apico-basal axes. The coordination of these individual cell shape changes has been shown to induce large tissue rearrangements[1–5].

As for tissue cellular organization, the apical surface of cells has been assumed to be a faithful proxy for their three-dimensional (3D) shape. Consequently, epithelial cells have been depicted as prisms with polygonal apical and basal faces. For example, during tissue invagination processes, such as the *Drosophila* mesoderm furrowing or vertebrate neurulation, epithelial cells change their shape from columnar to the so-called "bottle" form[1–3]. When schematized, the bottle shape is pictured as a variation of a prism, the frustum, i.e., the portion of a pyramid that remains between two parallel planes[6]. Frusta display apical and basal polygonal faces with the same number of sides but with a different area[1–3]. Thus, it is generally assumed that the cell organization in the apical surface drives the epithelial 3D architecture.

The arrangement of cells in the apical surface of the epithelium has been extensively analysed from biophysical, mechanical, and topological viewpoints[1,7–16]. These studies have been essential to understand fundamental morphogenetic processes, such as convergent extension, tissue size and shape control, and organogenesis. Topologically, the apical surface of epithelial sheets is arranged similarly to Voronoi diagrams. The Voronoi formalism has been shown to be useful to understand the mechanisms underlying tissue organization in the plane of the epithelium[7,17]. Moreover, any curved surface and 3D structure can be partitioned by means of Voronoi cells using computational geometry tools[18–20].

Several groups have tried to go beyond the two-dimensional description of tissues combining computational models and experimental systems[21,22]. This has been done by analysing the apical surface of 3D structures[23,24] or by developing lateral vertex models to study epithelial invaginations[25,26]. Recently, studies have focused on understanding 3D curved epithelia[27,28]. Khan et al. quantified epithelial folding by tracking individual cells during *Drosophila* gastrulation and showed intercalations in the plane of the epithelium and shape changes[29]. Other studies have addressed the emergence of curved 3D structures (e.g., tubes and spheroids) by means of numerical simulations[3,21,22,30–38]. Notably, in all these works epithelial cells are, anew, described and modelled as either prisms or frusta. However, there is evidence that epithelial cells are able to contact different neighbouring cells at different depths along the apico-basal axis of the cell (contrary to the prism/frustum paradigm). The appearance of these intercalations along the apical-basal axis has been observed in the columnar epithelium of *Drosophila* imaginal discs[39] or during *Drosophila* germ-band extension[40,41] and has been also modelled computationally in the context of a planar tissue[42].

Altogether, there is a gap of knowledge about the 3D packing of epithelial cells in curved tissues and, by extension, about the associated morphogenetic processes that create these structures. In addition to this fundamental aspect of morphogenesis, the ability to engineer tissues and organs in future critically relies on the ability to understand, and then control, the 3D organization of cells[43,44]. Here we combine a theoretical/computational framework with experimental data to quantify and characterize the 3D structure of curved epithelia. Remarkably, our approach unveils new aspects of cells and their packing properties. Thus, in curved tissues such as tubes, vaults, or spheroids, epithelial cells adopt a previously undescribed geometrical shape that makes their packing energetically efficient. Here we propose that such geometrical conformation is one of nature's solutions to epithelial bending.

## Results

**A tubular model reveals apico-basal cell intercalations**. To investigate whether 3D packing in curved epithelia could be simply explained by prismatic cells adopting a frusta-like shape (Fig. 1a, b), we computed a cylindrical epithelium that mimics the architecture of epithelial tubes and glands (Methods section, Fig. 1c). In this model, the ratio between the outer cylinder radius ($R_b$, basal radius) and the inner cylinder radius ($R_a$, apical radius) determines how large the relative expansion of the basal surface is with respect to the apical surface. We called this magnitude the surface ratio of the tube: $R_b/R_a$ (Fig. 1c). The model assumes that apical and basal surfaces behave as Voronoi diagrams. A Voronoi diagram is generated by a set of seeds in space. A Voronoi cell is then defined by the unique region of space that contains all points closer to a given seed than to any other. First, we generated a Voronoi diagram on the inner surface (apical). Then, the seed of each Voronoi cell was projected to the closest point on the surface of an outer cylinder (basal) generating a second Voronoi diagram in the basal surface of the tube. Interestingly, the Voronoi diagrams that emerged from apical and basal surfaces showed different topologies: their cells had different neighbours on each surface (Fig. 1c). This change was due to the asymmetric enlargement of the area between basal and apical surfaces that caused the redistribution of the distances between the corresponding seeds. The distance between two neighbouring seeds along the longitudinal direction of the tube was the same in apical and basal surfaces (Fig. 1c, green and red cells). On the contrary, the distance between neighbouring seeds along the transverse direction increased on the basal surface (Fig. 1c, blue and yellow cells).

This topological argument entails a fundamental consequence: a change in the identity of neighbouring cells from the apical to the basal surface is not compatible with the view of epithelial cells as either prisms or frusta (Fig. 1a, b). In fact, in a four-cell motif (Fig. 1c), the neighbour exchange means a cell intercalation process along the apico-basal axis. This spatial transition would be analogous to the temporal T1 transition leading to cell intercalations in a number of morphogenetic events, such as convergent extension[14,45,46]. Therefore, in this manuscript, we will refer to this topological change in four-cell motifs as apico-basal intercalations or apico-basal transitions.

We then analysed in detail the theoretical geometrical shapes or solids participating in this topological transition. The exchange of neighbours involved the appearance of a vertex along the apico-basal axis of these solids, at the location where the transition takes place (Fig. 1d). To the best of our knowledge, this type of geometric figures had been previously undescribed. We coin the term scutoid to name this geometrical solid for its resemblance with the scutum/scutellum shapes in the thorax of some insects such as the *Cetoniidae* subfamily of beetles (Fig. 1e). Scutoids, in contrast to prisms/frusta, are not necessarily convex, and present non-planar lateral faces (Fig. 1d, f, g, and Methods). This is due to the fact that the boundary between cells follow geodesic trajectories (the shortest path connecting two points)[20]. Importantly, this ensures that scutoids display concave surfaces that allow the 3D packing of cells in a curved tissue (Fig. 1d, f, g).

**Scutoids appear in tubular epithelia**. Our model predicts that scutoids should exist in actual curved epithelia. To determine this, we examined the third larval instar salivary gland of *Drosophila*, a model for tubulogenesis that is extensively used, where cubic

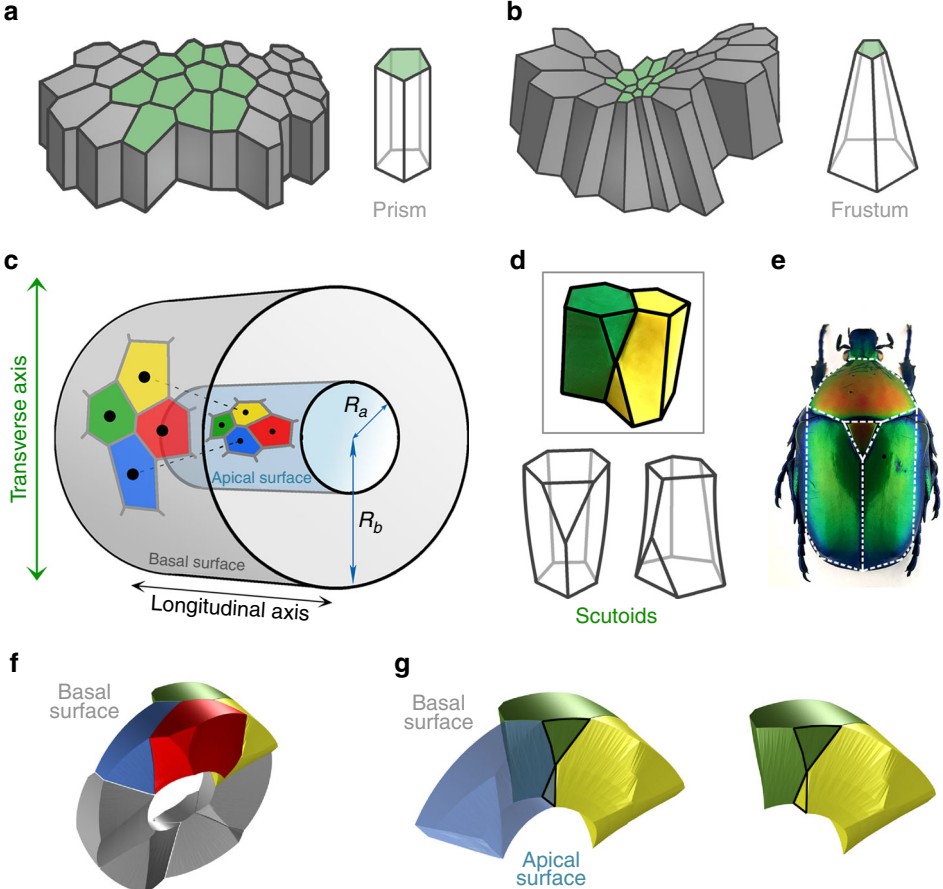

**Fig. 1** A mathematical model for curved epithelia uncovers a novel geometrical solid. **a** Scheme representing planar columnar/cubic monolayer epithelia. Cells are simplified as prisms. **b** Scheme illustrating an invagination or fold in a columnar/cubic monolayer epithelium. Cells adopt the called "bottle shape" that would be simplified as frusta. **c** Mathematical model for an epithelial tube. A Voronoi diagram is drawn on the surface of a cylinder (representing the apical surface of the epithelial tube). The seeds of each Voronoi cells are projected in an outer cylinder (representing the basal surface of the epithelial tube). This can induce a topological change, a cell intercalation. Yellow and blue cells are neighbours in the apical surface but not in the basal surface. The reciprocal occurs for red and green cells. $R_a$, radius from the centre of the cylinders to the apical surface. $R_b$, radius from the centre of the cylinders to the basal surface. **d** Modelling clay figures illustrating two scutoids participating in a transition and two schemes for scutoids solids. Scutoids are characterized by having at least a vertex in a different plane to the two bases and present curved surfaces. **e** A dorsal view of a *Protaetia speciose* beetle of the Cetoniidae family. The white lines highlight the resemblance of its scutum, scutellum and wings with the shape of the scutoids. Illustration from Dr. Nicolas Gompel, with permission. **f** 3D reconstruction of the cells forming a tube with $R_b/R_a = 2.5$. The four-cell motif (green, yellow, blue, and red cells) shows an apico-basal cell intercalation. **g** Detail of the apico-basal transition, showing how the blue and yellow cells contact in the apical part, but not in the basal part. The figure also shows that scutoids present concave surfaces

secretory cells arrange around a lumen to form a cylinder[47,48] (Fig. 2a). We developed a method to identify and track in depth (i.e. along the apico-basal axis) the epithelial cells in this gland (Fig. 2a, Supplementary Movie 1, Methods). In agreement with our predictions, we identified numerous apico-basal transitions (716 cells examined, 75 ± 17% of scutoids, surface ratio: 7 ± 3, Methods and Fig. 2b). Using confocal images, the 3D reconstruction of cell volumes confirmed the appearance of "scutoidal" shapes (Figs. 1g, 2c).

To assess additional geometrical features underlying the emergence of apico-basal transitions, we exploited further the predictive capabilities of the Voronoi tubular model. (Fig. 3a, Supplementary Fig. 1 and Methods). First, we computed nine exterior expansions of the tube to span a wide range of surface ratio conditions. We obtained a series of increasing surface ratios with the formula $1/(1 − x)$ taking $x$ the values {0.1, 0.2, …, 0.9} (Fig. 3a, b and Methods). The expansion of the exterior face of the tube caused the basal Voronoi cells to increase proportionally their area, to tile the surface (Fig. 3a). As a baseline, we obtained

the model for the planar epithelium $\left(\frac{R_b}{R_a} = 1\right)$ that represented the case without a topological change between the two surfaces (Supplementary Fig. 1). At the same time, we examined the influence of cell density by testing a different number of cells (40, 80, 200, 400, and 800 cells) on the same surface (Fig. 3a, b and Methods). We quantified the percentage of cells presenting at least one apico-basal transition (Fig. 3b and Supplementary Data 1). Such percentage increased as a function of the basal surface enlargement in all the cases, reaching 100% when $\frac{R_b}{R_a} = 10$ (Fig. 3b). The increment rate was similar for the models with 80, 200, 400, and 800 cells, and slightly smaller for 40 cells. In addition, we measured the number of transitions per cell. The expansion of the basal surface produced an increasing number of cells participating in several intercalations. Again, the numbers were lower in the case of 40 cells (Supplementary Fig. 2 and Supplementary Data 1). These results confirmed that the change of surface ratio, leading to variations of the cellular aspect ratio, is the main contributor to the emergence of apico-basal transitions

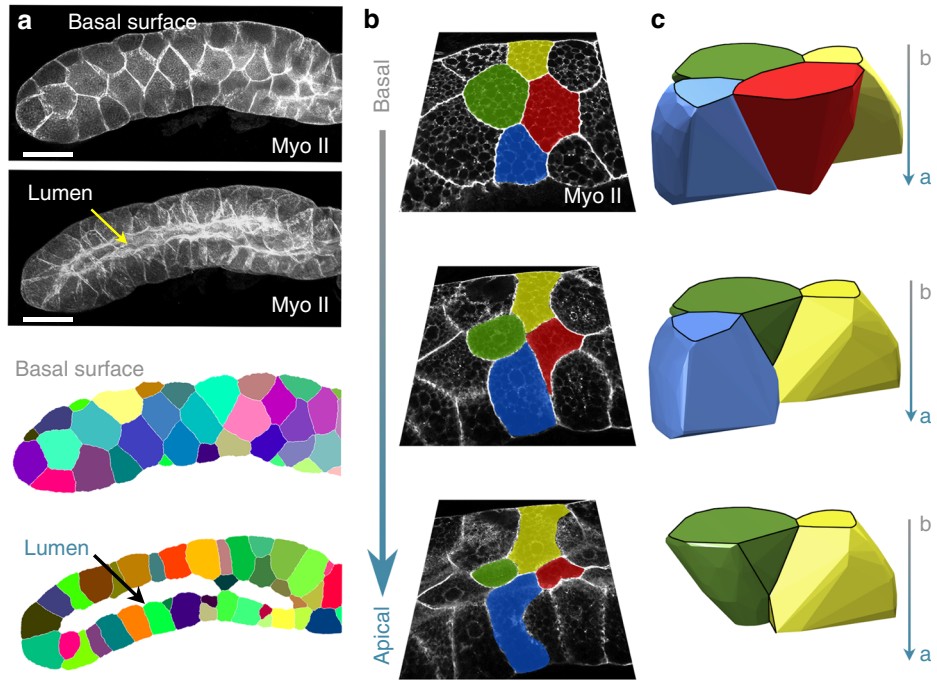

**Fig. 2** 3D tissue packing of curved epithelia. **a** Example of *Drosophila* salivary gland and its processed images. Scale bar = 100 μm. **b** Confocal images showing the apico-basal cell intercalation of epithelial cells marked with green, yellow, red, and blue pseudo-colours. The green cell participates in two apico-basal transitions. **c** 3D reconstruction of the same cells labelled in **b** using the same colour code. The image confirms the presence of concave surfaces predicted by the mathematical model

and that cell density affects the appearance of scutoids in a minor way.

Since the apico-basal intercalations necessarily imply changes in the edge shared by cells, we also investigated the geometrical traits of four-cell motifs. Note that the aforementioned edge has to change its size and rotate during an apico-basal transition (Fig. 3c and Methods). To that end, we first collected information from four-cells motifs, with and without apico-basal transitions ($n = 40$ larval salivary glands, 146 motifs with transitions, 656 motifs without transitions, Methods). The measurements were taken using the confocal images of the outer (basal) surface of the glands. We observed that the angle of the edge between the contacting cells was smaller than 30° in most of the transitions analysed; in contrast, the no transitions cases showed a much wider range of angles (Fig. 3d). In addition, we found that the length of the edge was shorter in four-cell motifs presenting apico-basal intercalations (Fig. 3d and Supplementary Data 2). As expected, we obtained equivalent results in the corresponding Voronoi tubes that show a similar percentage of scutoids (1/0.6 surface ratio) (Fig. 3e). In this regard, the analysis of the different Voronoi tubes revealed two clear results: (i) for small surface ratios the transitions appeared mainly when the edge angle was between 0° and 15° with respect to the transverse axis of the cylinder (measured in the basal surface); (ii) an increase of the surface ratio led the motifs with transitions to have larger angles and edge lengths (Supplementary Fig. 2). In all these cases, there were significant statistical differences between transitions and no transitions regarding edge angle and edge length distributions (Kolmogorov–Smirnov test; Supplementary Data 1) indicating that in tubular geometries, the appearance of apico-basal transitions is not random but associated to certain geometrical traits of the four-cell motifs.

**A line-tension minimization framework to analyse 3D packing.** The geometric characteristics of the solids that appear due to the apico-basal cell transitions suggest the existence of a

minimization principle governing the cellular packing motifs. From that perspective, we extended the theoretical model to analyse the biophysical origin and consequences of the appearance of the apico-basal transitions (Fig. 4a, b). Given that the latter is ultimately spatial T1 transitions, our approach focuses on the minimal model able to generate neighbour exchanges. In that regard, it has been shown that line-tension energy variations, i.e., the energetic changes due to the remodelling of the membrane shared by cells in a given apico-basal plane, are necessary and sufficient to induce T1 transitions[49] (Discussion).

Thus, a cell packing configuration in any plane perpendicular to the basal-apical direction is circumscribed to a quadrilateral (Fig. 4a). Our theoretical model relies on an idealized situation where packing is described in terms of two limiting stable configurations that are characterized by the length of their edges, $l_w$ and $l_h$. Here, for simplicity, we also use $l_w$ and $l_h$ to refer to the packing configurations. Therefore, when estimating the energy from experimental measurements (Methods), we assumed that an actual configuration can be effectively represented as a linear combination of those fundamental modes (Fig. 4a). In addition, a third, unstable, configuration is possible if $l_w = l_h = 0$, (fourfold vertex configuration, Fig. 4a, b). By focusing on the tensile forces controlling cell interactions, an energy functional can be defined[35,46,50–52]. Under these conditions, the energy landscape of a particular motif can be determined, and the stability of packing configurations can then be ascertained as a function of the length of the edge and the cellular aspect ratio, see Fig. 4b (Methods). We note that the changes in the aspect ratio, $\epsilon = \frac{h}{w}$, as a readout of the surface ratio anisotropy, relies on the assumption that cellular motifs are not affected by neighbouring configurations.

According to our Voronoi model, the larger the surface ratio of a tube, the more scutoids develop (Fig. 3b). Also, our data indicate that the shorter the edge length, the more likely it is that a transition appears (Fig. 3d, e). These two phenomena can be qualitatively explained by our line-tension minimization

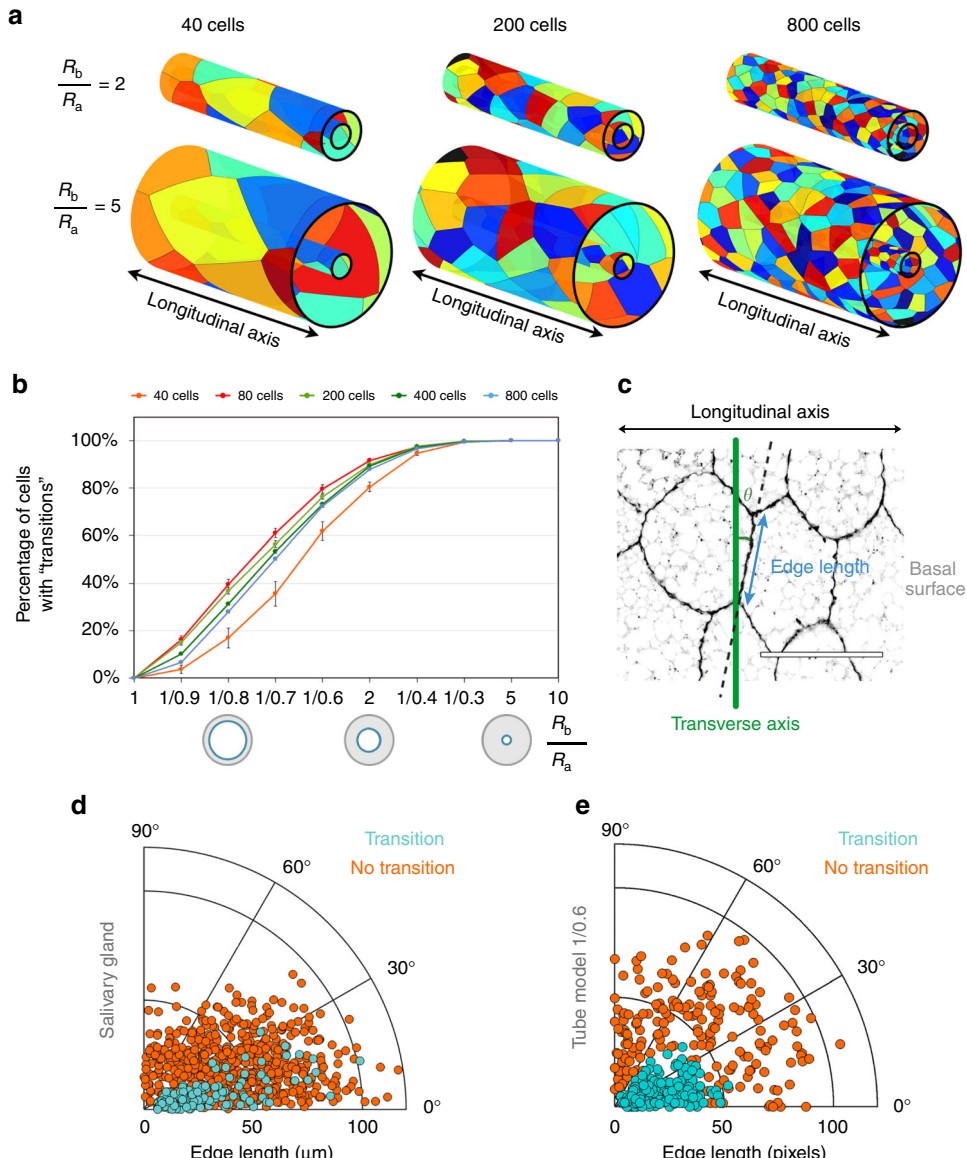

**Fig. 3** Apico-basal transitions are favoured by geometrical factors. **a** Examples of the 40, 200, and 800 cells models resulting from basal expansions of two and five times the apical surface. Data are represented as mean ± SEM. **b** Percentage of cells involved in transitions (scutoids) in relation to the increase of the surface ratio for five conditions (40, 80, 200, 400, and 800 cells). **c** Scheme showing how are measured the edge angle with respect the transverse axis ($\theta$) and the edge length in a four-cells motif of the basal surface of the salivary gland. Scale bar = 100 μm. **d** Polar scatter showing the length and the angle of the contacting edge in basal four-cell motifs from the salivary gland epithelium. Light blue points stand for motifs that exchange neighbours; orange points stand for motifs that do not intercalate. **e** Polar scatter showing the length and the angle of the contacting edge in basal four-cell motifs from the tubular model with the 1/0.6 surface ratio. Light blue points stand for 200 motifs that exchange neighbours; orange points stand for 200 motifs that do not intercalate

framework since, on the one hand, the energy landscape is further explored in terms of the aspect ratio (surface ratio anisotropy) and, on the other hand, for a given tube, scutoids ($l_w \leftrightarrow l_h$ transitions) are favoured when, close to the unstable fourfold vertex configuration, the surface ratio anisotropy enables jumping above the energy barrier (Fig. 4b and Methods). The experimental edge configurations and the values spanned by the aspect ratio of cellular motifs leading to no transitions or transitions in Voronoi tubes and salivary glands are in agreement with our energetic considerations about these data (Fig. 4c, Supplementary Fig. 3, and Methods). This suggests an analysis of transitions and no transitions from the viewpoint of the path followed in the energy landscape to

explore the attractors and enable energy relaxation. That path can be effectively characterized by basal-apical trajectories in the $E_h$, $E_w$ energy space (Methods), thus accounting for the change of the weight of each fundamental packing configuration as one moves along the apico-basal axis of the cell. We computed the energy of cell configurations in the apical and basal surfaces in salivary glands and Voronoi tubes (Fig. 4d, e, Methods). In the case of no transitions (frustra packing), we obtained short, disordered, local rearrangements corresponding to fluctuations around the same energetic attractor. On the other hand, in the case of transitions, we found long trajectories due to jumps over the fourfold vertex energy barrier to reach a new attractor and achieve further stability

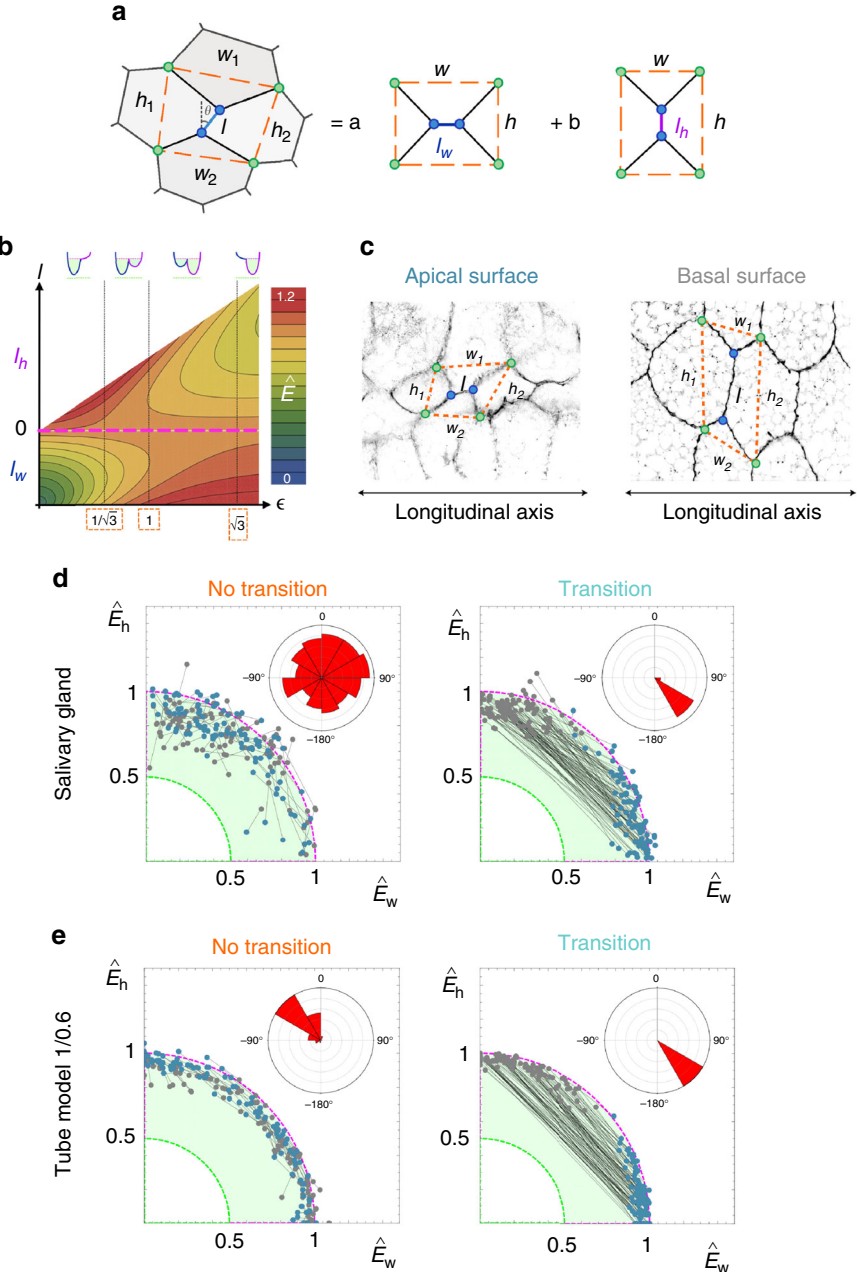

**Fig. 4** Energy minimization on tubular epithelia. **a** Packing configurations are characterized by four-cells motifs. Our theoretical argument to explain packing relies on line-tension energy minimization. Two idealized, stable, packing configurations are possible along the apico-basal axis as represented by $l_w$ and $l_w$. We decompose the experimental measured value of the line-tension energy into these fundamental modes. **b** Density plot of the energy profile (dimensionless) as a function of the aspect ratio, $\epsilon = h/w$, and $l$ (in units of $w$) as obtained by the theoretical model. If $\epsilon < 1/\sqrt{3}$ the only stable configuration is $l_w$, if $\epsilon > \sqrt{3}$ the only stable configuration is $l_h$, and if $\epsilon \in (1/\sqrt{3}, \sqrt{3})$ both configurations are stable. Cells adopt the "scutoidal" shape when there is a $l_w \leftrightarrow l_h$ transition. The dashed magenta line indicates the location of the unstable configuration $l_w = l_h = 0$. The potential wells (top) indicate schematically the shape of the energy profile within each zone: the green-shaded regions indicate the stable energy attractors and the green dotted line the absolute energy minimum (see **d**, **e**). **c** Scheme showing how the experimental values for $h$, $w$, and $l$ were measured in the apical and basal surfaces. Four-cell motifs were identified and we characterized the aspect ratio $\langle h \rangle / \langle w \rangle$ by measuring $\langle w \rangle = (w_1 + w_2)/2$ and $\langle h \rangle = (h_1 + h_2)/2$. **d–e** Decomposition of the tensile energies into the fundamental modes $\hat{E}_w$ and $\hat{E}_h$ (Methods) for apico-basal events in salivary glands (**d**) and 1/0.6 Voronoi tubes (**e**): "no transition" (left) "transition" (right) events. Individual packing configuration are represented by connected dots that account for the energy at the basal (grey) and apical (blue) surfaces. The magenta and green dotted lines indicate the theoretical energy levels of the unstable fourfold configuration and the absolute energy minimum respectively. The stable energy attractors are located within the green-shaded region. Insets: polar histogram accounting for the directionality of the trajectories from basal to apical

(Fig. 4b, d, e). Interestingly, these trajectories showed a clear directionality as one moves from the basal to the apical surface of the tissue (from $l_h$ to $l_w$) in agreement with what is expected from the theoretical energy landscape.

**Scutoids are a general feature of curved epithelia.** Our model predicts that apico-basal cell intercalations may appear whenever there is an asymmetric enlargement of the basal and apical areas. Thus, we predicted that the appearance of scutoids in curved

epithelia should not to be an exclusive of the larval salivary gland but rather, a general cornerstone of epithelial architecture. In this regard, we analysed the epithelial folds in the *Drosophila* developing embryo (4 embryos analysed, 214 cells examined, $50 \pm 15\%$ of scutoids, surface ratio: $1.6 \pm 0.2$, Methods and Supplementary Fig. 4a–c). We also selected, for the same embryos, 5 regions of interest at locations without folds and a smaller surface ratio $(1.2 \pm 0.1)$. In this case, the number of scutoids decreased (734 cells examined, $15 \pm 4\%$ of scutoids, Methods). These results confirmed, in an independent context, a correlation between surface ratio anisotropy and the appearance of scutoids in curved epithelia.

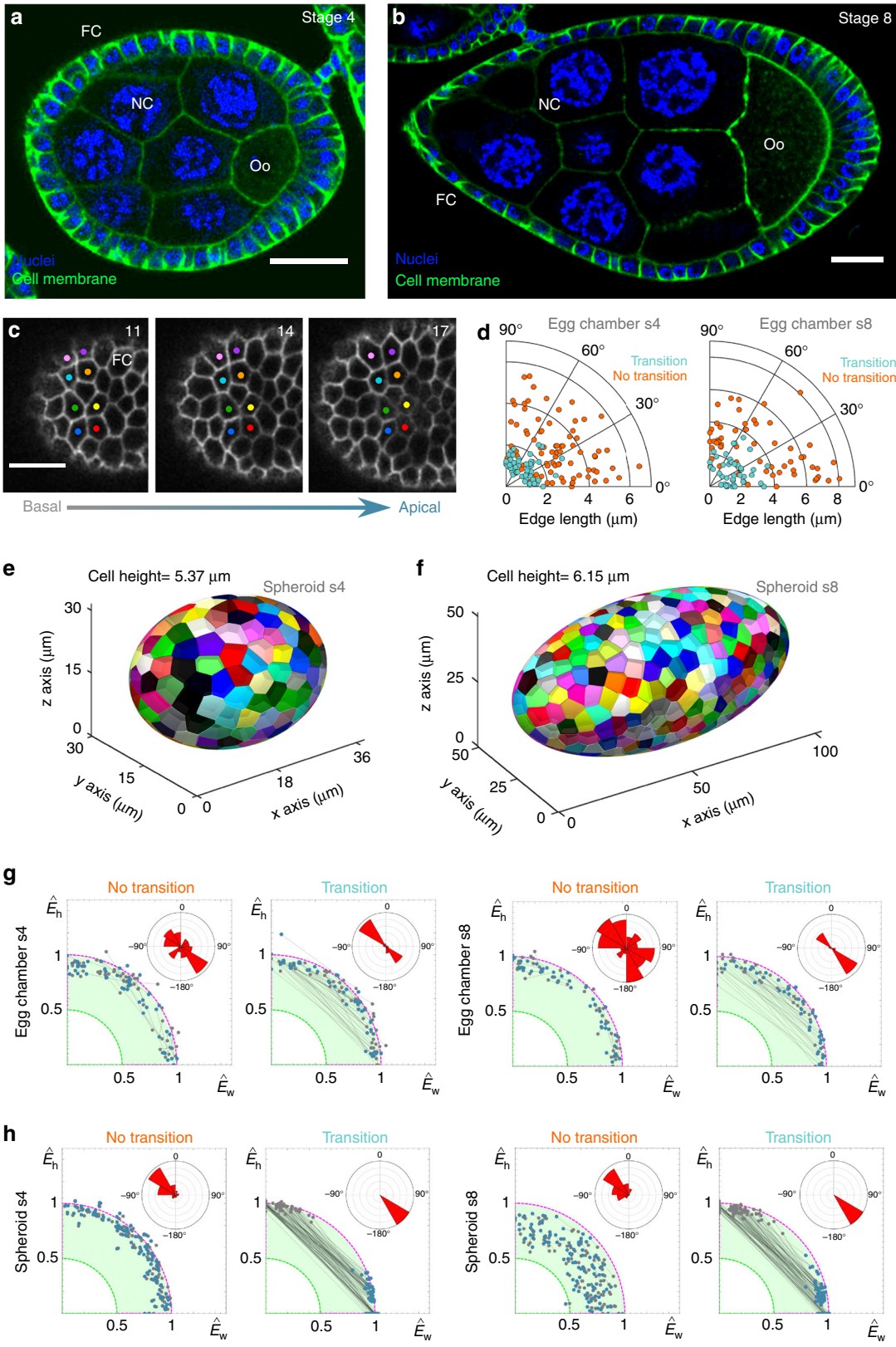

A tubular geometry, or a fold, is a particular case where one of the main curvatures is null, whereas the other is constant[53]. To extend our conclusions to a general framework, we explored other curved epithelia that have non-null curvatures along their symmetry axes (closer to a sphere or a spheroid). The *Drosophila* egg chamber is a spheroidal developmental structure, where a monolayer of follicle cells surrounds a group of germ cells. During development, the egg chamber elongates from a quasi-spherical shape to an apparent spheroidal form. This dynamic process is orchestrated through a collective migration of the follicular cells[54–56]. We obtained and processed confocal stacks from stage 4 (mild elongation) and stage 8 (complete elongation) egg chambers (Fig. 5a, b, Methods) and the data revealed the presence of apico-basal transitions in the follicular epithelium at both stages (Fig. 5c and Methods). In addition, we found that the percentage of scutoids decreased during the egg chamber maturation process (stage 4, 13 egg chambers analysed, 719 cells examined, 20 ± 8% scutoids; stage 8, 14 egg chambers analysed, 1283 cells examined, 10 ± 3% scutoids, Methods). This raises interesting questions about the importance of non-stationary processes during development (Discussion).

Contrary to the larval salivary glands, but likewise the locations without folds of the *Drosophila* embryo (Supplementary Fig. 4a–c, f and Supplementary Data 2), we did not detect a preferential orientation of the edge angle in four-cells motifs leading to apico-basal transitions in egg chambers (Fig. 5d, Supplementary Data 2 and Methods). This was also the case in the quasi-spherical multilayered Zebrafish epithelium at 50% epiboly stage, where while we registered apico-basal cell intercalations (7 embryos analysed, 6724 cells examined, 2.9 ± 1.5% of scutoids, Supplementary Fig. 4d, e, Supplementary Data 2 and Methods), the orientation of the edge of four-cells motifs did not display a clear bias (Supplementary Fig. 4g and Supplementary Data 2).

**A Voronoi model for spheroidal epithelia**. The aforementioned experimental evidence suggests that scutoids are a general feature of many curved epithelia. In order to further investigate the 3D packing in tissues that have non-null curvatures along both symmetry axes, we implemented a Voronoi spheroidal model (Methods). We note that in the particular case of the sphere, the basal to apical surface reduction is isotropic. Consequently, there is no change of the aspect ratio and the cells are expected to have the frusta shape[53].

First, we tested the appearance of scutoids under different combinations of the length of the spheroid radii and cell apico-basal sizes (Supplementary Fig. 5a–c and Supplementary Table 2). We observed that the increase in the percentage of scutoids correlated positively with the cell apico-basal length and with the ratio of the length of the axes. However, in the case of the sphere (i.e., when the length of the symmetry axes matched), we did not observe any scutoid regardless of the apico-basal length of the cell (Supplementary Fig. 5a and Supplementary Table 2). These data support our energetic model, where apico-basal transitions are driven by the anisotropy of the main curvatures ratios

between both surfaces, which ultimately leads to changes in the aspect ratio.

We computationally mimicked stage 4 and 8 egg chambers by using the measured average values of the symmetry axes lengths and the cell apico-basal size (Fig. 5e, f, Supplementary Data 3 and Methods). In the case of the Voronoi spheroidal model stage 4, the percentage of scutoids registered was smaller than in its actual tissue (130/15,285 simulations/cells analysed, 6 ± 4% scutoids). This percentage increased in the Voronoi spheroidal model stage 8 (30/7051 simulations/cells analysed, 11 ± 4% scutoids, Methods). It is worth to notice that in the actual egg chamber the number of scutoids decreased as this developmental structure became more prolate (i.e., a spheroid elongated towards the poles) from stage 4 to stage 8 (see Discussion). The second difference with respect to the actual egg chambers was the appearance of the bias, due to the differential curvature, in the orientation of the "transitions" as in tubular geometries (see Discussion, Supplementary Fig. 5d, e and Supplementary Table 1).

Finally, we computed the edge configurations and the values spanned by the aspect ratio of cellular motifs and also the energetic trajectories in the case of the transitions and no transitions for both the Voronoi spheroidal model and the egg chambers (Fig. 5g–h and Supplementary Fig. 3c–f). The results of our analyses were compatible with those in the case of tubular geometries, including the long trajectories in the case of transition packing and the local fluctuating character of no transition packing. Interestingly, the directionality of the trajectories in the case of transitions in the actual egg chambers suggested a role of active developmental processes in the 3D tissue packing (see Discussion). Altogether, our results support the hypothesis that the surface ratio anisotropy is able to drive energetic transitions to provide tissue stability, and therefore, scutoids facilitate 3D packing of curved epithelia.

## Discussion

Throughout embryogenesis, epithelia undergo dramatic morphogenetic changes that involve tissue bending and invagination. In general, the formation of any curved epithelium requires a fundamental 3D tiling adjustment: the same cells have to pave two surfaces that could vary largely in area[57]. Until now, this was envisioned to occur by a reduction on the surface area of the cells in just one side: i.e., an apical (or basal) constriction that confers a bottle shape to the cells[1–3,58] (Fig. 1a, b). Our results rule out this possibility as a general explanation of tissue packing. Instead, we propose a model in which curved epithelia are not formed just by prisms or frusta when a certain surface ratio asymmetry is exceeded (Fig. 6a, b). This prediction is supported by the identification of cell intercalations along the cellular apico-basal axis in diverse curved epithelia (Figs. 2 and 5, Supplementary Fig. 4). Therefore, our results reveal that curved tissue packing relies on another solution that manages cell shape and packing: apico-basal intercalations that give rise to scutoids. These previously undescribed geometric figures are not necessarily convex solids and

---

**Fig. 5** Graphic summary of 3D packing in epithelia. **a** Section of a stage 4 egg chamber with nuclei (blue) and cell membranes (green) labelled. The green staining decorates the contours of the follicular cells (FC), the nurse cells (NC) and the oocyte (Oo). Scale bar = 20 μm. **b** Section of a stage 8 egg chamber with nuclei (blue) and cell membranes (green) labelled. The green staining decorates the contours of the follicular cells (FC), the nurse cells (NC) and the oocyte (Oo). Scale bar = 20 μm. **c** Close up of the surface of a stage 4 egg chamber (**a**), showing two four-cell motifs that are involved in an apico-basal transition. The numbers indicate the confocal plane, increasing from basal to apical regions of the follicular cells. In this case, the sections were taken every 0.4 μm. Scale bar = 10 μm. **d** Polar scatters showing the length and the angle of the contacting edge in basal four-cell motifs from the stage 4 and stage 8 egg chambers. Light blue points stand for motifs that exchange neighbours; orange points stand for motifs that do not intercalate. **e** Voronoi spheroid model with the same dimensions that the average egg chamber 4. **f** Voronoi spheroid model with the same dimensions that the average egg chamber 8. **g–h** Linear decomposition of the experimental energies into the fundamental modes $\hat{E}_w$ and $\hat{E}_h$. Colour codes as in Fig. 4

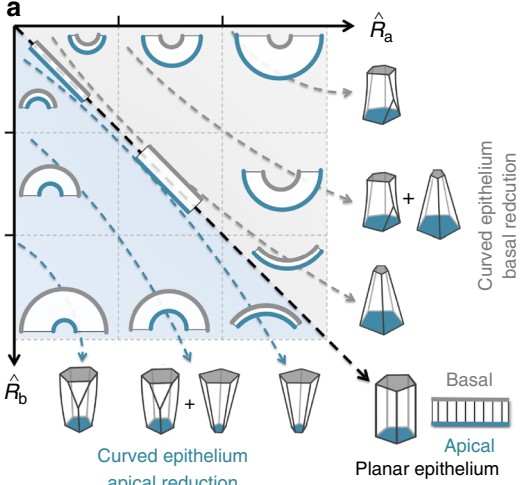

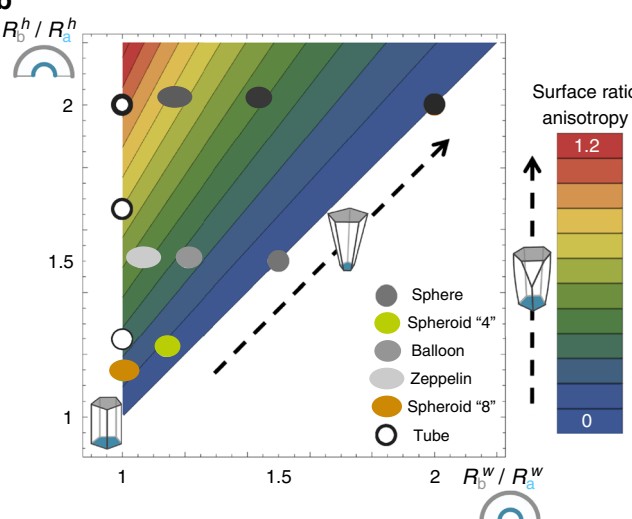

**Fig. 6** Graphic summary of 3D packing in epithelia. **a** The graph represents the curvature ratio (surface ratio) depending of the dimensionless values of $\hat{R}_a$ and $\hat{R}_b$. The curvature ratio is defined by the ratio $\hat{R}_a/\hat{R}_b$ in the case of basal reduction and $\hat{R}_b/\hat{R}_a$ in the case of apical reduction. The bottom triangle (blue shaded region) covers the region where the epithelial bend generating a reduction of the apical surface $\hat{R}_b > \hat{R}_a$. The top triangle (grey shaded region) shows the region where a reduction of the basal surface occurs $\hat{R}_a > \hat{R}_b$. In all the geometrical solids and epithelial representations basal is at top (grey) and apical is at bottom (blue). The black dashed arrow in the main diagonal indicates the epithelium without curvature (ratio = 1). This planar epithelium is formed by prisms. The packing configuration space is compartmentalized into regions depending on the relative values of the curvature radii (or equivalently the aspect ratio) of epithelia. The curved arrows indicate the types of solids that are more favourable in the epithelia depending of the curvature: close to the diagonal the frusta shapes of the cells are preferred. On the contrary, in the top-right and bottom-left corners all the cells tend to be scutoids that take part in several apico-basal transitions. There is also an intermediate situation where the epithelium is packed using both scutoids and frusta shapes. **b** The model can be generalized to describe tissues with two main axes of curvature (h, transversal, and w, longitudinal), where R represents the Cauchy radius, e.g. $R_b^h$ accounts for the Cauchy radius along the h axis in the basal, b, surface. Here we assume that the apical surface lies in the inner part of the tissue. As a function of the Cauchy radii it is possible to assess the surface ratio anisotropy for different geometries as measured by the relative change of the aspect ratio from apical to basal (colour code): $(R_b^h/R_a^h)/(R_b^w/R_a^w) - 1$. The increment of the surface ratio anisotropy positively correlates with the percentage of scutoids

stage 4 egg chamber, the percentage of scutoids is larger in the epithelium than in the Voronoi model. Moreover, in stage 4 follicular epithelium, we do not find a relationship between the orientation of the four-cells motifs and the appearance of apico-basal transitions (Fig. 5d). From stage 1 to 8, the follicular epithelium of the egg chamber is a very dynamic tissue that is rotating due to a coordinated migration that induces the elongation of the whole structure[54]. These morphogenetic events are accompanied by an increase in size mediated by cell proliferation that ceases at stage 6. Consequently, in this context, our analyses were performed under highly dynamic conditions in which the tissues had not yet adopted their final morphology (contrary to the stationary condition in the larval salivary gland). Interestingly, it has been recently shown how epithelial cells can form migrating basolateral protrusions during cell intercalation in the *Drosophila* germ-band extension process[40]. These protrusions would initiate the tissue rearrangement, and imply, like scutoids, a different topology in the apical and basal surfaces. Thus, we propose that the observed differences are a consequence of the active cell and developmental processes. This hypothesis is supported by the double directionality of the energetic trajectories in the actual egg chambers (compare insets in Fig. 5g, h). In contrast, in the case of Voronoi spheroids, the trajectories are clearly driven by energetic cues following the most stable state as a function of the surface ratio anisotropy. In general, we reckon that 3D epithelial packing is a multifactorial process and that the plasticity of the epithelial cells in some conditions (e.g., long columnar cells) will be coordinated with the geometrical constraints to accommodate cells in the tissue. All these findings open new promising directions, yet outside of the scope of this study.

Tissue bending implies the difference in size between the apical and basal surfaces of epithelial cells, which in turn induces the emergence of apico-basal intercalations. As we mentioned above, apico-basal cell intercalations are analogous to the well-known T1

allow the accommodation between concave and convex surfaces in the apico-basal transitions. In our model, we typically observe that contacts between the cells can be concave (Fig. 1g). This feature has been observed in diverse epithelia[59] and is very clear on the basal surface of the larval salivary gland (Fig. 2b, c).

Our Voronoi tubular model predicts and explains the appearance of the apico-basal transitions and key topological features. We have found that the increment of the surface ratio positively correlates with the percentage of scutoids (Fig. 3b). This behaviour was also observed in the *Drosophila* larval salivary glands and in the *Drosophila* embryo folds. The agreement between the results obtained in the Voronoi tubular models and in these tissues also includes the preferential orientation of the motifs that are involved in an apico-basal transition (Fig. 3d, e). Yet, we acknowledge limitations to explain all experimental data by our modelling approach. For a given surface ratio, a larger number of scutoids develops in the Voronoi tubular model compared to the case of the larval salivary gland. We argue that these differences could be explained by the biophysical characteristics of the cells, such as their stiffness or surface stress, that are neglected in the Voronoi model. In practice, cellular reorganization must actively overcome energy barriers whereas a purely geometrical description assumes an idealized (friction-free) scenario.

The comparison between the egg chambers and the spheroid models brings very interesting conclusions. In the case of the

transitions[14,45,46] where the role played by time in the latter is played by space (apico-basal radial coordinate) in the former (Fig. 2b). T1 transitions have been shown to enable cellular rearrangements that stabilize tissues[60,61]. In this context, our tensile energetic model recapitulates previous results showing that changes in the cellular aspect ratio promote cellular rearrangement[62]. Here we propose that scutoids help to stabilize 3D packing in curved epithelia. To explore this idea our model focuses on the minimal energetic considerations that can induce neighbour reorganization: tensile forces. As in the case of T1 transitions, turgor pressure, surface stress, and the contractility of the cortex (including the effect of the acto-myosin ring at the apical junctional complex in epithelial cells) could certainly contribute to shaping the conditions under which a scutoid develop. Moreover, our energetic model considers isolated cellular environments (i.e., configurations) thus disregarding the effect of the surrounding cells. Notwithstanding the fact that experimentally characterizing these additional energetic contributions is challenging, our results support the premise that the coupling between geometrical changes due to bending and line-tension energy variations is a fundamental driving force in scutoid formation.

Additionally, our model facilitates a general intuition to understand 3D packing in epithelia. In the simple case of a tube, this can be understood in terms of the surface ratio (the relation between the dimensionless radii $\hat{R}_a$ and $\hat{R}_b$); (Fig. 6a). When $\hat{R}_b/\hat{R}_a = 1$, apical and basal surfaces present the same area, the epithelium is flat and formed by prisms. If $\hat{R}_b/\hat{R}_a \approx 1$, i.e., similar but not equal, like in the case of a tiny tube formed by a squamous epithelium (top-left corner), the differences between apical and basal surfaces are small and the whole tissue is formed by frusta. Note that this sort of epithelial packing occurs as long as $\hat{R}_a$ and $\hat{R}_b$ increase simultaneously (Fig. 6a, top-left to bottom-right diagonal). On the contrary, when $\hat{R}_a \gg \hat{R}_b$ or vice-versa (top-right corner and bottom-left corner) our model predicts that the tissue will be exclusively formed by scutoids (Fig. 6a). In intermediate situations, our model predicts different proportions of frusta and scutoids (Fig. 6a).

In the case of geometries involving two non-null principal curvatures (e.g. spheroids), the same arguments apply: the anisotropy of the surface ratio is able to drive energetic transitions (Fig. 6b). In spheres, where there is not any change in the cellular aspect ratio as in planar epithelia, the appearance of scutoids due to tensile forces is not expected. Also, in tissues with a spheroidal geometry (e.g., egg chamber), we would expect that the more prolate the shape of the tissue, the larger the percentage of scutoids. Finally, the model also considers tubes as a limiting case where one of the curvatures is null. In that situation, the effect of the surface ratio increase over the percentage of scutoids is more pronounced.

In this work, we provide a more accurate and detailed view of epithelial 3D packing. It now becomes important to investigate how this new framework affects distinct processes during development and disease, from the morphogenesis of organs to the environment at the onset of tumour formation[1,11,50,63]. Altogether, our study paves the way to understand the biomechanics of morphogenesis in developing organisms and sheds light on the underlying logic of 3D cellular organization. This is fundamental not only for an understanding of tissue architecture during development and disease, but to the fields of tissue and organ engineering[43].

## Methods
**Geometry of Voronoi diagrams in curved surfaces**. From the mathematical point of view, in order to model a tissue, we defined: the space, the metric, and,

finally, the seeds. With these ingredients, we built a Voronoi diagram, whose cells will be a model for the epithelial cells in a curved tissue (Fig. 1c).

Regarding the space, we started with a given surface $S$, then for each point $X$ $(u,v) \in S$, there exists a vector $N(u,v)$ of length 1 that is orthogonal to the tangent plane in $X(u,v)$ to $S$. Thus, for each $\lambda \in [0,1]$, it is possible to define a new surface $S_\lambda$ parallel to $S$ in such a way that any point of $S_\lambda$ is $X_\lambda(u,v) = X(u,v) + \lambda N(u,v)$ (a point in one of the surfaces has an equivalent point in each one of the parallel surfaces).

The metric in each one of the surfaces previously defined is just the length of the shortest geodesic joining two points. In the case of the cylinder, the geodesics are helices in the cylinder. In the case of the sphere, the geodesics are arcs of great-circles.

We defined every seed starting in a point on the apical surface. That point defined a segment between the basal and the apical surfaces by means of its normal (given the point $X(u,v)$, the segment is $X(u,v) + \lambda N(u,v)$, $\lambda \in [0,1]$). The intersection of these line segments with a given surface determined a seed. Thus, in order to generate all the seeds, in a first step we have chosen $n$ points on the apical surface, then the $n$ segments that were generated by them, and, finally, the intersection of those segments with every surface $S_\lambda$ defined the seeds for that surface.

The next step was to compute the Voronoi diagrams of the seeds obtained in each one of the parallel surfaces. We linked the Voronoi regions corresponding to seeds on the same segment obtaining a 3D figure. In some cases, we obtained figures with vertices not located neither in apical nor in basal surfaces. These solids are not prisms or frusta (Fig. 1f, g). Moreover, it must be pointed out that this geometrical figure does not need to be convex.

**Immunohistochemistry and confocal imaging of the epithelia**. Flies were grown at 25 °C by employing standard culture techniques. Zebrafish were maintained and bred according to standard procedures[64]. For each type of sample different immunohistochemistry conditions were used.

For *Drosophila* larval salivary glands we used the *sqh-GFP* line[65] and Cy3-labeled phalloidin (Sigma) to label the cell contours of the epithelial cells. Salivary glands from third instar larvae were dissected in PBS and fixed with 4% paraformaldehyde in PBS for 20 min. The samples were washed three times for 10 min with PBT (PBS, 0.3% Triton) before phalloidin incubation for 1 h. Larval salivary glands were mounted using Fluoromount-G (Southern Biotech) using double sided tape to avoid the squashing of sample[66]. Images were taken with a Nikon Eclipse Ti-E laser scanning confocal microscope. The images were captured using ×20 dry (for the complete glands that were segmented) and ×40 immersion objectives and exported as a 1024 × 1024 pixels TIFF file.

For *Drosophila* embryos, embryos collected for 6–8 h after egg laying were dechorionated and fixed in 1:1 formaldehyde 4% in PBS:n-heptane for 20 min at room temperature and stained according to standard protocols. For basolateral membrane staining embryos were incubated at room temperature 4 h with anti-Disc large antibody (1:200 in PBS—0.1% Tween—0.1% BSA; mouse, Developmental studies hybridome bank, 4F3). Secondary antibody was coupled to Alexa488 (Molecular Probes, 1:500). Images were taken on a SPE Leica confocal microscope equipped with a ×20 or ×40 ACS-APO oil objective and processed using FIJI and Adobe Photoshop CS6.

For *Drosophila* egg chambers, to label the follicle cells membrane the ubiquitously expressed membrane marker Resille-GFP (II) was used[67]. For ovaries staining 1–2-day-old females were fattened on yeast for 48 h before dissection. Ovaries were dissected in PBT (phosphate-buffered saline + 0.1% Tween 20). Fixation was performed incubating egg chambers with 4% paraformaldehyde in PBS on ice for 20 min. For nuclei labelling fixed ovaries were incubated with Hoechst (Molecular Probes ™) 1:1000 in PBT. The antibody goat anti-GFP-FITC (abcam, ab6662) was used 1:500. Individual ovarioles without the muscle sheath were mounted in Vectashield (Vector Laboratories). Egg chamber images were captured with a Leica TCS-SPE confocal equipped with a ×40 (1.15 NA) ACS-APO oil objective.

For Zebrafish embryos membranes and nuclei of zebrafish embryos were labelled in vivo by microinjecting a bicistronic mRNA (GAP43-GFP:h2bRPF) at one-cell stage. mRNA for microinjection was synthesised using the mMessage Machine kit (Ambion), following manufacturer's instructions. Embryos were grown up to pre-gastrula stage, fixed in 4% paraformaldehyde and embedded in 1.5% low-melting point agarose (as in ref. [68] for imaging under an inverted confocal microscope "NIKON A1R+ in vivo", using a 20X/0.75 dry objective. Images were processed using FIJI/ImageJ. All procedures comply with the guidelines from the European Community and Spanish legislation for the experimental use of animals.

***Drosophila* larval salivary gland measurements**. In order to understand the cellular organization in a 3D salivary gland, we measured several geometrical properties from selected motifs. Importantly, to estimate these measurements we assumed that the glands have a cylindrical shape. These motifs were chosen considering the four-cells that were contacting alongside all the sections. Furthermore, at least two of these cells had to be central cells: a central cell was defined as a cell situated at the middle of the gland and its whole area was completely visible from basal to apical. As a rule of thumb, we counted as a valid motif when there were no fourfold vertex configurations in any of the two surfaces. We applied this consensus

to all the images from actual tissues. Once the valid motifs were selected, we measured four properties related to epithelial architecture, directly in the confocal stack. We took as a reference the central edge that two cells of the motif were sharing in the basal (outer) surface (Fig. 3c). They were the surface ratio of the gland, the central edge length, the central edge angle of the motif, and the percentage of central cells with "scutoidal" shape

Surface ratio estimation: We measured the average width of the lumen ($2R_a$) and the average width of the salivary gland ($2R_b$) at the region of the selected motif by using the FIJI's Straight Line tool. The surface ratio was then calculated as $R_b/R_a$.

Edge angle and length: We defined the angle of the motif's intercellular edge, which is the edge shared by cells that have three neighbours within the motif, and the orientation it had with respect to the transverse axis. We quantified the edge angle and the edge length using FIJI's angle and FIJI's straight-line tools respectively. (Fig. 3c, d).

Percentage of scutoids: We quantified the percentage of central cells participating in at least one transition along the apico-basal axis using all the sections of the confocal stack.

In addition, three geometric properties in the salivary glands images were measured to feed the line-tension minimization model (Supplementary Data 3). They were the sum of all the edges length into the motif packing configuration ($L_T$), width ($w$) and height ($h$).

Total length of edges ($L_T$): We gauged and summed the length of every intercellular edge in the packing configuration from each four-cells motif (Fig. 4a, c). We used FIJI's straight-line tool.

Width ($w$) and height ($h$): We measured the "width" and "height" from each four-cells motif in basal and apical surfaces. The "width"/"height" is defined as the distance between two vertices in the quadrilateral that defines a four-cells configuration along the longitudinal/transverse axis (Fig. 4a, c). The average "width" $\langle w \rangle$ and the average "height" $\langle h \rangle$ of a motif were calculated as the average of each measurement: $w_1$ and $w_2$, and $h_1$ and $h_2$ (Fig. 4a, c), which were estimated using FIJI straight-line tool.

**_Drosophila_ embryo measurements**. We analysed confocal stacks imaging the whole length of the _Drosophila_ embryo epithelial during gastrulation stage. First, we selected two types of regions of interest (ROI), one corresponding to the flatter embryo surface, and a second one corresponding to the embryonic folds (Supplementary Fig. 4a–c). For each one of these ROIs, we counted the total number of cells and the percentage of scutoids using FIJI.

In the folds, we measured edge length and edge angle in four-cells motifs taking as a reference the basal surface. We chose ten four-cells motifs (at most) where "transition" occurred and another ten with "no transition". The edge angle and length were measured similarly to the larval salivary gland (Supplementary Fig. 4f). The surface ratio was calculated by measuring the area of the four-cells motif in basal (using FIJI's polygon selection tool) and dividing it by the area of the four-cells motif in apical. The same procedure was followed with the ROIs corresponding to the flatter surface of the embryo (Supplementary Fig. 4f). In this case, edge length and edge angle in four-cells motifs were measured on the apical surface. Then, we calculated the surface ratio by dividing the apical area by the basal area (since in this case, the epithelium presents a basal constriction).

**_Drosophila_ egg chamber measurements**. To further investigate the presence of scutoids in _Drosophila_ egg chamber, we have measured several geometric aspects of motifs with a transition and with no transition in both apical and basal layers. In particular, we calculated the value for the parameters $w_1$, $w_2$, $h_1$ and $h_2$ and $L_T$ (see line-tension minimization model) along with edge angle and length (Fig. 4a, c), following the same procedure from the larval salivary gland. These motifs were captured from a visible central region of interest (ROI), discarding the cells from the borders of the egg chamber. We took all the follicular cells that appeared just before any nurse cell was completely visible. Moreover, we manually counted the number of cells that were found on the defined ROI and the total number of cells that appeared previous to the polar cells. We also estimated the number of scutoids (cells with at least a scutoidal side) and the cells which were not scutoids.

In order to estimate the geometrical shape of the egg chamber and its influence in the scutoids appearance, we measured the radii of the $X$-axis, in anterior and posterior, and the $Y$-axis. Furthermore, to appraise the mean cell height of each sample, we measured the height of several cells that were located, laterally, on the region of the transverse axis, where the curvature was maximum.

**Zebrafish embryo measurements**. We inspected the first two layers of cells from zebrafish embryo samples at 50% epiboly. We first captured at most 10 cellular motifs with apico-basal transitions and 10 with no transition. To further analysed them, we segmented the four-cell motifs in their apical and basal frames, capturing the following measurements: edge angle and length regarding the $X$-axis and surface ratio estimation (getting the ratio between the area of the motif in apical and in basal) (Supplementary Fig. d, e, g). We also counted the number of cells in the first two layers, along with the percentage of scutoids.

**Voronoi tubular model**. We have designed a geometric model to simulate curved epithelia. The model was based on generating a Voronoi diagram in a cylindrical

shape. This process required that Voronoi cells located in an extreme of abscissa axis (the transverse axis of the cylinder image) were continuous in another abscissa extreme[20] (Supplementary Fig. 1). We have used Matlab R2014b (Mathworks) as our computational tool. Taking as a reference the previous 2D Voronoi model established in[7], we implemented a Centroidal Voronoi Tessellation (CVT) on the lateral surface of the cylinder. To this end, we developed the cylinder lateral surface in the Euclidean plane and applied Lloyd's algorithm to a random Voronoi diagram[69,70].

These are the detailed steps for the method: (a) We randomly located a finite number of seeds (40, 80, 200, 400, and 800) in a defined Euclidean space (512 × 4096 pixels) for 20 different images. (b) We tripled the images with all the seeds in the direction of cylindrical transverse axis (abscissas axis), getting an image with size 1536 × 4096 pixels, and the number of seeds tripled (c) Voronoi regions were delimited by proximity over all the seeds, producing a Voronoi diagram. (d) The position of all the Voronoi cell centroids was calculated, and the centroids located between in the central 513 and 1024 abscissas coordinates were stored for next steps. (e) Images were cropped, capturing the [513-1024] x [1-4096] pixels interval, giving as result 512 × 4096 pixels images. (f) The procedure from (a) to (e) was repeated using the calculated centroids as new seeds in (a) to perform Lloyd's algorithm. This algorithm was repeated four times for each image, achieving 5 diagrams for each initial random image.

To study the tubular model, we chose the Voronoi 5 diagram of the CVT, since it resembled a homogeneous distribution of the cells[7]. The 20 Voronoi 5 diagrams that represented the cylinder were used to create a new lateral surface of a new cylinder with a larger radius and the same height (Fig. 3a and Supplementary Fig. 1). In this way, we were able to generate a model that represented the inner (initial) and the outer (expansion) lateral surfaces of a tube. The surface expansion was established in 9 steps using the following formula: $f(x) = 1/(1 - x/10), I \subset \mathbb{N}, I = [1, 9], \quad \forall x \in I : 1 \leq x \leq 9$. The calculations were made by maintaining fixed the $R_a$ length (inner surface) and increasing proportionally the $R_b$ length (outer surface, Supplementary Fig. 1).

To partition the outer surface with Voronoi cells we used the following strategy: we multiplied the size of abscissas axis of the original image (apical) by the surface ratio ($R_b/R_a$) (Supplementary Fig. 1). In a similar way, abscissas coordinates ($X$) of the Voronoi seeds were multiplied by the same factor, getting a new position of the seeds in the basal surface that corresponds to the given surface ratio expansion. Using these seeds, we applied again (b), (c), and (e) steps, using the size of the enlarged image. The Voronoi cells corresponding to the original seed and their projections were tracked and labelled with a specific colour allowing their identification along different surfaces. (Figs. 1f–g and 3a and Supplementary Fig. 1).

**Voronoi tubular model measurements**. The Voronoi tubular model was used to measure the edge length and the edge angle of all the intercellular edges (Fig. 3c). These calculations were made on the basal surface and repeated for each one of the nine surface ratios. The angles were calculated taking as reference the transverse axis of the cylinder image (abscissas axis of a planar image). For each one of these intercellular edges, we established their involvement in apico-basal transitions (Fig. 3b, Supplementary Fig. 2a and Supplementary Data 1) and obtained the percentage of scutoids. We discarded the edges shared by non-valid cells (cells located at the tube tip), and we only quantified scutoids along the set of valid cells (not located at the tube tip in any surface, apical or basal). Finally, 200 random measurements were taken from transition and no transition edges to represent the scatter polar graphics (angle vs edge length) (Fig. 3e and Supplementary Fig. 2b–e).

Additionally, a set of geometric properties were quantified to feed the line-tension minimization model (Supplementary Data 3): the sum of all the edges length within the motif packing configuration ($L_T$), width ($w$) and height ($h$) (Fig. 4a, c).

To calculate the mentioned parameters, we captured the vertices involved in the packing configuration of four-cells motifs and we measured the distances between them. The four-cells motifs used in our analysis kept their four-cells together in apical and basal surfaces and we discarded the motifs that did not satisfy this requirement. We also disregarded the motifs with non-valid cells on any of these surfaces and we classified our measurements: four-cells motifs with and without apico-basal transition. Similar to the process followed in the actual tissue images, and in order to avoid artefacts, we discarded the motifs presenting fourfold vertex configuration in one of the two surfaces. To make this process automatic, we only analysed the motifs with an edge length longer than 4 pixels. This number was selected after our analyses in the computed Voronoi spheres, where the basal to apical surface reduction is isotropic and all cells must have frusta shape[53]. However, in the cases of four-cell motifs with very short edges, we found false positives (apico-basal transitions) in the sphere models. According to our analyses, 4 pixels set the minimum threshold for which false positives were removed. For consistency, we applied this condition to both tubular and spheroid models.

First, we got all the possible energetic measurements that satisfied out restrictions, discriminating between transition and no transition motifs. These datasets were used to represent the cellular motifs' aspect ratio (Supplementary Fig. 3b). Second, we chose 100 samples of energetic measurements for both types of four-cells motifs: we randomly selected, at most, 10 measurements (5 from transition edges and 5 from no transition edges) from each cylindrical Voronoi Diagram (20 different diagrams). In summary, we took a maximum of 100

measurements in transition edges and 100 in no transition edges, for each surface ratio. This data set was used to map the energetic trajectories from basal to apical (Fig. 4e).

**Voronoi spheroidal model.** A 3D Voronoi model was developed simulating the behaviour of a curved epithelium with two non-null principal curvatures. First, we implemented spheroidal models mimicking the *Drosophila* egg chamber in the stages 4 and 8 of development. Second, we explored several spheroidal structures (in terms of its radii and apico-basal length of the cells) in order to study the cell packing with different degrees of curvature. The inputs of the Voronoi spheroidal model were its three axial radii, number of cells (N) and apico-basal length of the cells.

We developed a pipeline to create different spheroidal shapes in terms of its axial radii, keeping constant *Y* and *Z* radii, while modifying *X*. By defining the axial radii *Y* and *Z* of the inner spheroid layer as the unitary value, we got the following spheroids: 10 random realizations of a sphere (*X* radius inner layer = 1), a balloon-like spheroid (*X* radius inner layer = 1.5) and a more prolate spheroid, 'Zeppelin'-shaped, (*X* radius inner layer = 2), with three different apico-basal length of the cells 0.5, 1 and 2 (Supplementary Fig. 5a–c); 130 random realizations of spheroid stage 4 (*X* radius inner layer = 1.26, cell height = 0.22, N = 200) and 30 random realizations of spheroid stage 8 (*X* radius inner layer = 2.16, cell height = 0.15, N = 450) (Fig. 5e, f), whose dimensions are proportional to the measurements taken in the *Drosophila* egg chamber at stage 4 and stage 8, respectively.

The pipeline followed these steps: (1) We put N seeds on the apical surface with a minimum distance between them (depending on the surface area), this distance avoided the overlapping and homogenized the seeds distribution along the ellipsoid. Note that N represents the maximum number of seeds because, in some cases, the surface could not be filled by all the seeds and satisfy the condition about the minimum distance between them simultaneously. (2) We extrapolated the seeds to the basal surface using the ellipsoid equation:

$$\frac{x}{a} + \frac{y}{b} + \frac{z}{c} = 1, \quad (1)$$

where parameters are: *a*, *b*, and *c* represent the radii along the *X*, *Y*, and *Z* axes of the ellipsoid respectively, and *x*, *y*, and *z* correspond to the *X*, *Y*, and *Z* coordinates in the ellipsoid surface, respectively. the case of the spheroid, *b* = *c*. (3) We determined the segments connecting the seeds in apical and their corresponding ones in basal. (4) We implemented the Voronoi algorithm, in the 3D space, taking as seeds the mentioned apico-basal segments to computed the Voronoi cells. (5) Finally, we applied again the ellipsoid equation, capturing only the surfaces of interest (apical and basal).

**Voronoi spheroidal model measurements.** The Voronoi spheroidal model was used to extract similar measurements that those taken in the Voronoi tubular model. Due to the impossibility to measure accurately in the 3D space, we took some measurements from the spheroid in an analogous way that was carried out for the confocal images of *Drosophila* egg chambers. This procedure is explained in detail below.

To transform each 3D spheroid into four 2D images, we computed two maximum intensity projections along the *Z*-axis (first from *Z* = zRadius to *Z* = 0, and second from *Z* = zRadius to *Z* = 0) and two others along the *Y*-axis (first from *Y* = yRadius to *Y* = 0, and second from *Y* = −yRadius to *Y* = 0) of each layer (outer and inner). After the projections were created, we homogenized the cell's border, eroding the cells and transforming their border to a width of one pixel. This procedure led to images similar to the ones obtained with the confocal microscope. Afterwards, we chose a region of interest (ROI) like the ones used in the *Drosophila* egg chambers measurements. This ROI excluded the cells from the border of the spheroid projections and only selected as valid cells, those in the central region (approximately from $X = -\frac{2}{3} * x\text{Radius}$ to $X = \frac{2}{3} * x\text{Radius}$). We then selected all the four-cells motifs that composed the ROI and measured the same parameters analysed in the Voronoi tubular model: the intercellular edge angle (with respect to the transverse axis of the ellipsoid projection), the intercellular edge length, the percentage of scutoids (Supplementary Table 2), the sum of the edges length ($L_T$), the width (*w*) and the height (*h*) of the four-cells motifs (Fig. 4a, c). The energetic measurements followed exactly the same constraints that we applied to the Voronoi tubular model: we chose the cellular motifs present in both apical and basal layers, excluding the non-valid motifs (containing non-valid cells). We also distinguished between transition and no transition motifs. As explained above, we only selected four-cell motifs with the edge length longer than 4 pixels (see Voronoi tubular model measurements section). Furthermore, we extracted two sets of data: one with all the possible motifs used for plotting the cellular motifs' aspect ratio; and another composed by 100 samples (selected randomly) to illustrate the energetic trajectories from basal to apical (Fig. 5h). To represent the geometric traits into polar scatter graphs, we randomly selected 200 edge length and angle measurements from transition edges and 200,200 from no transition edges (Supplementary Fig. 5d, e).

Additionally, we measured the maximum and minimum curvatures of the 3D ellipsoid coordinates in the inner and outer surfaces[53]. We analysed the curvatures from $X = -\frac{2}{3} * x\text{Radius}$ to $X = \frac{2}{3} * x\text{Radius}$, covering the remaining axis entirely and matching the ROI of the projected images.

**Kolmogorov–Smirnov statistical test.** We performed a two-sample Kolmogorov–Smirnov decision test to check if two samples came from the same continuous distribution (Supplementary Table 1). In particular, we analysed whether the edge angle and the edge length had different distributions in the motifs leading to transitions and to no transition. For this purpose, we have used the Matlab function 'kstest2', using as source data the same measurements represented in the polar scatters plots (Figs. 3d, e, 5d, Supplementary Fig. 2b–e, Supplementary Fig. 4f–g and Supplementary Fig. 5d–e).

**3D segmentation of *Drosophila* larval salivary glands.** We designed a protocol to segment, identify and track every epithelial cell in the confocal images taken from the larval salivary gland (Fig. 2a). In all the cases, the images were confocal stacks containing the cells contours along the height of the epithelial cells. The protocol had several steps: (1) Segmentation: We used Trainable Weka Segmentation[71] (FIJI) a machine learning-based plugin. We labelled two categories: the cells' outlines, and the inner part of the cells. The outlines included the perimeter of the cells and the lumen. The inner part included the body of the cells and the rest of the image. After the training step, we applied the plugin to all the planes of the confocal stack, getting a first set of segmented images. (2) Manual correction: this step was necessary to reflect rigorously the real cell outlines in the segmented images. This was especially important in the segmentation of the lumen of the larval salivary gland. The process was performed using Adobe Photoshop CS6. (3) Image homogenization: we developed a Matlab R2014b (Mathworks) script to execute and accomplish a final post-processing to equalize the size of the cells' border in order to improve the segmented images. (4) 3D segmentation through the confocal stack: we built a semi-automatic method that tracked the cells throughout plane *Z* of the confocal stack. This script, which was developed in Matlab, integrated the information from the positions and the areas of the cells at different stacks. Despite these calculations, it needed an additional step of human curation. As final output, we obtained segmented stacks where every cell was labelled (Fig. 2a and Supplementary Movie 1).

**3D reconstructions.** Using the data from the mathematical model, we performed the 3D reconstruction of a four-cell motif to visualize the surfaces of the cells along the apico-basal axis. We used the Voronoi cylinder images with different surface ratios as the source to generate the Voronoi tube. We applied the cylinder Matlab function to get a 3D representation of the whole model. We chose the original image that represented the inner part of the tube, a basal expansion image with surface ratio = 2.5, and we additionally built the intermediate layers between these surfaces. Cells were identified in each layer and we used Matlab's alphaShape function to calculate the tightest fitting shape for each cell, producing a volume that embodied a 3D Voronoi cell (Fig. 1f, g). To represent the Voronoi tubular simulations with different surfaces ratios as a real 3D cylinder, we made use of Matlab's function 'wrap' (Fig. 3a).

Using a similar approach, we obtained a 3D cell representation from motifs located at the *Drosophila* salivary gland. We selected a cellular motif from a segmented stack of images (see above). In order to model the 3D shape of each cell, first, we captured the boundary pixels of a cell in all the layers. Since the source images were obtained from plane confocal sections, we included a correction step where *Z* coordinates were extrapolated to a cylinder using an inferred cylinder radius for each layer. Once the cell boundaries were obtained, we defined the convex hull of the points which shaped the cell using boundary Matlab function. Finally, we reconstructed the 3D shape using the surface Matlab function (Fig. 2c). To illustrate the shape the spheroids that we analysed, and the contained cells, we performed a 3D reconstruction (Fig. 5e, f and Supplementary Fig. 5a–c). For this purpose, we have used the function 'alphaShape' to get the shape of each cell. The source data being the coordinates we got from applying the Voronoi algorithm to the initial seed segments thus representing cells' height.

**Line-tension minimization model.** The existence of tensile forces controlling cell interactions, either driven by cadherin and/or acto-myosin activity, can be described through the following line-tension energy functional:

$$
\begin{aligned}
E \;=\; & \sigma\left[l_w + 2\sqrt{(w - l_w)^2 + h^2}\right] H(l_w)H(w - l_w) \\
& + \sigma\left[l_h + 2\sqrt{(h - l_h)^2 + w^2}\right] H(l_h)H(h - l_h),
\end{aligned}
\quad (2)
$$

where $\sigma$ stands for the line tension, that we assume to be constant, and *H* for the Heaviside step function. Using *w* as a characteristic length scale and the energy of the unstable (fourfold vertex) configuration, $E_0 = 2\sigma\sqrt{h^2 + w^2}$, as an energetic reference, the dimensionless energy of a four-cell configuration reads,

$$
\hat{E} = \frac{\left[-l + 2\sqrt{(1+l)^2 + \epsilon^2}\right] H(-l)H(1+l) + \left[l + 2\sqrt{(h-l)^2 + 1}\right] H(l)H(h-l)}{2\sqrt{\epsilon^2 + 1}},
\quad (3)
$$

where $\epsilon \in (0, \infty)$ stands for the aspect ratio $h/w$ and $l \in (-1, \epsilon)$, such that the negative and positive values represent $l_w$ and $l_h$, respectively.

If dissipative forces are neglected, the relaxational dynamics of packing configurations due to tensile forces reads,

$$\frac{\partial l}{\partial t} = -\frac{\partial \hat{E}}{\partial l}, \qquad (4)$$

and stable configurations (force balance) are prescribed by the energy minima (Fig. 4b). As a function of the aspect ratio, different regimes can be distinguished: if $\epsilon \in \left(0, \frac{1}{\sqrt{3}}\right)$ there is a single global minimum located at $l < 0$, if $\epsilon \in \left(\frac{1}{\sqrt{3}}, \sqrt{3}\right)$ there are two global minima located (one at $l < 0$ and another one at $l > 0$); finally, if $\epsilon \in (\sqrt{3}, \infty)$ there is a single global minimum located at $l > 0$. Namely, if $\epsilon \in \left(0, \frac{1}{\sqrt{3}}\right)$ then the only stable configuration for packing is $l_w$ and if $\epsilon \in (\sqrt{3}, \infty)$ the only stable packing configuration is $l_h$. In case $\epsilon \in \left(\frac{1}{\sqrt{3}}, \sqrt{3}\right)$ both packing configurations are energetically stable.

On the one hand, no transition packing (i.e., frustra) is the only energetic stable configuration if along the radial coordinate between the apical and basal surfaces the range of variation of the aspect ratio is kept either smaller than $\frac{1}{\sqrt{3}}$ or larger than $\sqrt{3}$. On the other hand, transition packing (i.e., scutoids) is the only possible stable configuration if along the radial coordinate there is a change in the aspect ratio from a value smaller than $\frac{1}{\sqrt{3}}$ to a value larger than $\sqrt{3}$ (or vice-versa). Under other conditions in terms of the variation of the aspect ratio (as a readout of the surface ratio anisotropy), we expect a mix of packing configurations since, as a function of the radial coordinate, a transition between configurations, $l_w \leftrightarrow l_h$, can occur.

The experimental (in vivo or in silico) dimensionless values of the energy of a packing configuration was estimated by,

$$\hat{E}_{\text{exp}} = \frac{\hat{L}_T}{2\sqrt{1 + \langle \epsilon \rangle^2}} \qquad (5)$$

where $\hat{L}_T$ is the total length of the packing configuration (all edges defining the motif inside a parallelogram) in units of $\langle w \rangle$. The fundamental components of the energy, $\hat{E}_w$ and $\hat{E}_h$, are determined by a decomposition in terms of the director cosines,

$$\begin{aligned} \hat{E}_w &= \hat{E}_{\text{exp}} \sin\theta \\ \hat{E}_h &= \hat{E}_{\text{exp}} \cos\theta \end{aligned} \qquad (6)$$

such as $\hat{E}_{\text{exp}}^2 = \hat{E}_w^2 + \hat{E}_h^2$ and where $\theta$ is the measured angle formed by the transition edge. As for the value of the aspect ratio, we measured $\langle w \rangle = (w_1 + w_2)/2$ and $\langle h \rangle = (h_1 + h_2)/2$ (the average width and height of the configuration motifs, respectively) in the apical and basal surfaces (Fig. 4a).

In Supplementary Fig. 3, to characterize the experimental motifs in terms of $l$ (either $l_h$ or $l_w$) and $\epsilon$, we classify the edge configuration as a function of the value of the angle. If the latter is larger than 50 degrees we assumed an $l_w$ configuration, if smaller than 40 we assumed an $l_h$ configuration. The intermediate cases between 40 and 50 degrees were discarded in our analysis to avoid artefacts. The density plot was obtained estimating the probability density of couples, $(l, \epsilon)$, with $l$ normalized by the value of $\langle w \rangle$, and applying a Gaussian kernel to smooth the results.

**Code availability**. The code is available at: https://github.com/ComplexOrganizationOfLivingMatter/Epithelia3D/tree/master/InSilicoModels/paperCode.

**Data availability**. The authors declare that all relevant data supporting the findings of this study are available within the paper and its Supplementary information files. Additional data are upon request.

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

## Acknowledgements

M.D.M.-B. and A.V.-E. work was funded by the Ministerio Español de Ciencia y Tecnología (grants numbers BFU2013-48988-C2-1-P and BFU2016-80797 to M.D.M.-B.). A.V.-E. is supported by a FPI studentship (BES-2014-068850) from the Ministerio de Economía y Competitividad. S.S. is supported by grant of the MICINN/FEDER to J.C.-G. Hombría (BFU2016-76528-P). F.C. work is funded by Spanish Government Grant BFU2014-55918 and BBVA Foundation Personal Grant IN[16]_BBM_BAS_0078. J.B. acknowledges core funding at Lehigh University. L.M.E. and P.G.-G. are supported by the Ramón y Cajal program (PI13/01347); L.M.E., A.M., and C.G. work is funded by the Ministry of Economy, Industry, and Competitiveness grant BFU2016-74975-P co-funded by FEDER funds. P.V.-M. is supported by a contract co-funded by the Asociación Fundación Española contra el Cáncer and the Seville University. A.M.C. is supported by Fundación Pública Andaluza Progreso y Salud (Consejería de Salud), ref. PI-0033-2014. C.F. and A.T. is supported by a contract from Sistema Nacional de Garantía Juvenil and Programa Operativo de Empleo Juvenil 2014–2020. We are thankful to Dr. Nicolas Gompel for the picture of the Cetoniidae: *Protaetia (Potosia) speciose* to illustrate the term "scutoid". We thank Dr. Paco Martín, Dr. José López-Barneo, Dr. Alberto Pascual, Dr. Colin Adrain and Dr. Matthew Freeman for helpful comments and corrections to the manuscript.

## Author contributions

L.M.E. designed the study with help from A.M., C.G., and J.B. P.G.-G. and P.V.-M. wrote the software for the mathematical model and performed analyses. A.T., C.F., A.M.C., M.L., A.V.-E., M.B.-G., O.S.-P.-H., F.C., S.S., and M.D.M.-B., collaborated in the obtaining, measurement of parameters and analysis of the confocal images. All authors participated in the interpretation of results, discussions, and the development of the project. J.B. and L.M.E. wrote the manuscript with input from all authors.

## Additional information

**Competing interests:** The authors declare no competing interests.

