## [Peer Review File · Nature Communications]

Reviewers' comments:

Reviewer #1, an expert in epithelial morphogenesis (Remarks to the Author):

This paper from the groups of Buceta & Escudero reveals that apico-basal topological changes are expected in epithelia with differential curvature. The authors neologise the incomplete cell-cell interfaces involved as 'scutoids'. The paper starts by demonstrating the 3D phenomenon with the help of Voronoi tessellation on epithelial tubes, proceeds to show that scutoids are seen in three real curved epithelia, and concludes with a more formal physical approach to predict when scutoids are likely to appear, based on cell-cell interface length minimisation.

The paper makes a significant contribution to a question that has not yet been explicitly answered in the field: what is the 3D structure of cells in curved epithelia? This needs to be understood in order to understand 3D morphogenesis.

Overall the paper is interesting and potentially of interest to a high profile journal such as Nat Comms, though there are significant issues.

Major

1. In convergence-extension movements, such as axis extension in chick, zebrafish and *Drosophila*, new cell-cell contacts are actively made as a result of planar-polarised Myosin-II activity. It is widely accepted in this field that cells are likely to make new contacts at particular depths, not as prisms. For example, their ref 55 explicitly measures new contacts being made in the single-layered germ-band either on the apical or basal side, reflecting a long-standing question in this area. Hisao Honda has also modelled non-prismatic (scutoid-based) intercalations (*Dev Dyn* 2008 DOI: 10.1002/dvdy.21609 Fig. 3), though not explicitly in relation to curvature. Incomplete 3D cell-cell interfaces, or scutoids, are therefore not new. However, that they are predicted specifically because of the curvature of epithelia is to our knowledge new and interesting. Greater care is needed in not overemphasizing the novelty of their scutoids in various places – in the hyperbole of the abstract and in some confusion over the cause of scutoids they report (whether they are driven actively or the passive response to curvature).

2. The specific novelty they report is that epithelial curvature generates scutoids. However, the precise relationship between curvature and the likelihood of scutoids is incompletely presented. The property of curvature that makes scutoids likely is anisotropic curvature, not curvature itself – this is not made clear and their 'theoretical' and 'biophysical' models do not explore the relationship between different curvatures in orthogonal directions. The salivary gland is a nice example of an epithelial tube with curvature in one orientation only, but each of the other 3 tissue examples presented have curvature in both orientations. The zebrafish EVL wraps an essentially spherical embryo, where prisms would be predicted because curvature in orthogonal orientations would be equal. The *Drosophila* egg chamber and germ-band tissues have principal and secondary axes of curvature, with the latter non-zero. Their predictions of scutoids as a result of curvature needs to explicitly account for the combinations of orthogonal curvature. It is not sufficient to claim that 'some' scutoids will be found in curved epithelia in a paper in a high profile journal. The underlying relationship needs to be fully described. In some of the epithelia analysed (germ-band, EVL), active cell intercalation is occurring, which involves scutoids (e.g. ref 55). How do the authors propose to disentangle situations where scutoids due to curvature and active processes are involved?

3. It was not quite clear why both the Voronoi and biophysical approaches were needed. I agree that some minimisation of interface surface area is probably a sensible way to think about predicting when a transition is likely to occur in a 'passive' epithelium (i.e. one without basal or planar polarised Myosin, for example), as has been employed for the 'biophysical' approach. Since Voronoi tessellation minimises the total lengths of junctions, this approach is also 'biophysical' in the sense in which the authors use it, and wouldn't it do as good a job as their 'biophysical'?

approach? I suspect it would give very similar predictions. A vertex-based model would be another contender, where the perimeter and interface length terms would provide (effective) mechanical control over the frequency of transitions in a stretched sheet (as in Fig. S1). For example, nearly 2 decades ago, Chen & Brodland (2000, J Biomech. Eng.) in Fig. 3 explore parameter space for transitions in a stretched sheet (here also with orthogonal convergence, but the principle is good), much like Fig. S1. Can the authors better justify why both models were necessary, their relative advantages, what is a theoretical model that isn't biophysical, and discuss what alternative approaches would likely bring, such as vertex-based/finite element approaches. The paper reads as though fig 4 was added separately, while it would be better if the unique aspects of Voronoi and biophysical models could be interwoven.

4. In Fig 2 and associated Fig S1 the effect of cell number (10-400 cells) on "transitions" is surely nothing to do with cell density but due to edge effects? If done with periodic boundaries along the tube, or for longer tubes, the differences between the lines in Fig 1B would vanish? If this is correct, it doesn't inspire confidence in the formal aspects of the paper.

Minor

1. It was not at first obvious why the neologism 'scutoid' had been chosen, expecting it to describe the 3D shape of cells with apico-basal twists. It became clear that the term refers to the triangular cell-cell facets of cells whose topology changes between apical and basal, and these triangles are indeed reminiscent of the scutellum of a hemipteran. It would not be a bad idea to add a picture or dorsal diagram of a hemipteran (like

[https://en.wikipedia.org/wiki/Scutellum_\(insect_anatomy\)#/media/File:Heteroptera_morphology-d.svg](https://en.wikipedia.org/wiki/Scutellum_(insect_anatomy)#/media/File:Heteroptera_morphology-d.svg)) to show this explicitly in the paper (by Fig. 1E?).

2. In the abstract, the following words or phrases are either too strong or inappropriate: "general principle" too strong, "scholarly publications" avoid, "necessarily undergo" can undergo, "compelled to adopt" adopt. The second half of the final sentence is misplaced hyperbole.

3. The quality of prediction in Fig. 3E is poor (why no stats?). Would something like edge length/centroid-centroid length have discriminated better? Is this figure superseded by Fig. 4F?

4. The point of Fig 2D&E and the text that go with it are not clear, as they are not subsequently used – a formal relationship must exist between angles, interface lengths and likelihood of transition that would be more useful than simulated trends? The accompanying text could be more formal and precise.

5. P.8, line 15: this is not tracking in time, as would be expected from the word, so I would 'track in depth'. P.22, line 2: 'cell tacking' is 3D segmentation.

6. "Some", used in various places should be replaced with a quantity.

7. That scutoids can be concave: agreed, but differential pressure between neighbouring cells, as used in the force inference methods by Brodland et al, also generates concave interfaces, and these are quite common.

8. Fig S4C. A transverse view of the U-shaped furrow would help make sense of the panels shown, and to visualise the apical and basal curvatures.

9. Fig S4E. Not clear which is the surface orientation with greatest curvature, so not clear whether highlighted transitions are in line with predictions of differential curvature.

Reviewer #2, an expert in biological patterning and self organisation (Remarks to the Author):

The paper proposes a novel method of generating shapes of epithelial cells, based on Voronoi tessellations, that leads to more diverse and realistically looking forms of cells emerging in 3D packing of curved epithelia, compared with commonly used prisms or frusta. The attention is concentrated on the novel feature of spacial intercalation that involves the appearance of a vertex along the apico-basal axis of the cells.

The paper is original and important and deserves publication but the presentation needs to be improved.

The arrangement of the paper makes reading and understanding difficult. The meaning of apico-basal transition is not well defined in the text. It appears to be used interchangeably with apico-basal cell intercalation and becomes better understood only in the Discussion where it is characterised as a spacial T1 transition. The "computed nine conditions" are not listed explicitly. The authors present a long sequence of Results, with a plethora of rather obscure terminology, and statements which the reader is supposed to blindly believe, while the essential mechanical theory is hidden in the Materials and Methods part stuck in the end, and, unfortunately, is far from being adequate.

Some details, like technicalities of the construction of Voronoi diagrams, preparing samples, and imaging, can be indeed hidden in this Appendix, but the energy considerations, appearing in the very end, are crucial. Voronoi diagrams are just a formal device, and only mechanical and biological evidence can justify the stated preponderance of the novel cellular shapes.

The energy computations are the weakest point of the paper, and the expressions in this part lack rational justification. They are based on the notion of line tension, which is a marginal feature that is largely considered to be irrelevant or at least not fundamental in continuum mechanics. This analysis should involve surface and bulk energies, analogous to line and area energies in the theory of 2D tessellations. These energies are not easy to quantify, as they depend on biological factors, such as cell adhesion, actin stress, and osmotic pressure. This analysis may remain outside the scope of this paper, as comprehensive theory might be outside the domain of expertise of this group or require long time to develop. I will support publication if the energy analysis is omitted, provided the paper and Supplementary Material contain sufficient information for the analysis by interested researchers.

The paper has a number of linguistic problems:

- The term "scutoids" is obscure and not suggestive of the generated forms that do not look as scales. Perhaps, "prismoids" is a better fit.
- The terms "apical" and "basal" are interchanged relative to the common use; it would be proper to call the outer surface apical, and in Figs.1,3 the apical side is below and the basal is above.
- p.7: obscure expressions: "enrichment of values", "apparition of transitions" (appearance? emergence?)
- p.14: "contacts between the cells are not convex" - The authors apparently mean that the sign of the mean curvature varies along the contact surface.
- p.17: "The metric in each one of the surfaces ... is just the distance of the shortest geodesic joining two points" - this might be the way the authors define distances but it is not the standard definition of the metric, which is a tensor.
- p.5: a funny typo: To spam a wide range of basal expansions.

Reviewer #3, an expert in epithelial morphogenesis (Remarks to the Author):

This manuscript predicts the presence of a 'novel' 3D epithelial cell shape that the authors term 'scutoid'. Voronoi modelling of cell packing in tubular structures at both apical and basal surfaces predicts transition of cell contacts along the apical-basal axis, termed intercalation, leading to these more complex shapes. The authors then go on to analyse tubular tissues, segment cell shapes and show that indeed scutoids can be found in many tissues. In the second part of the manuscript they use a model of line tension and energy minimisation to predict that energetically a packing involving transitions or scutoids is also favourable.

A few general comments first: the most interesting aspect of the paper in my mind is that the authors show that in tubular tissues of a certain radius ratio, space filling packing of epithelial cells requires the appearance of complex shapes such as scutoids, but overall this paper does not clearly state the case as to why this should also be true for other systems of epithelial bending that are not cylindrical tubes.

Extensive rewriting and restructuring could make the manuscript more accessible and convincing.

Also, part of what the authors suggest has been observed qualitatively previously, i.e. intercalation along the apical-basal axis itself in epithelia is not novel and in fact has been described as early as 1991 by Condic, Fristrom and Fristrom (Apical cell shape changes during *Drosophila* imaginal leg disc elongation: a novel morphogenetic mechanism. *Development*, 111(1), 23–33.) in case of *Drosophila* leg discs. What the authors add here is the quantitative aspect.

The authors make a good case of examining the salivary gland example, but what I find somewhat confusing is why scutoids are also found in several of the other examples such as follicle cells in stage 4 egg chambers and the EVL during zebrafish epiboly. The geometric constraints of these tissues are very different to the theoretical voronoi cylinder as well as the salivary gland model described first, and this is not properly discussed. Especially in the case of epiboly, the overall shape is near spherical, i.e. similar amounts of curvature to the epithelium in all directions, whereas a cylinder by definition is curved only in one of two orthogonal directions. So would voronoi modelling in such case also predict scutoids? Scaling of configurations from apical to basal in a spherical case should suffice to lead to a space-filling epithelial packing. Isn't it more likely that the intercalations along the apical-basal length observed in the case of epiboly are consequences of the dynamic behaviour of the tissue, i.e. snap shots of events that occur during more complex transitions of dynamic epithelial tissues? The same could explain these intercalations found in the follicular cells surrounding the germ line, as these begin a highly dynamic behaviour of rotation around the germline at stage 4 that is shown here (Cetera, M., Juan, G. R. R.-S., Oakes, P. W., Lewellyn, L., Fairchild, M. J., Tanentzapf, G., et al. (1AD). Epithelial rotation promotes the global alignment of contractile actin bundles during *Drosophila* egg chamber elongation. *Nature Communications*, 5, 1–12. <http://doi.org/10.1038/ncomms6511>). Another consideration that should at least be discussed is that as a general point, I would argue that because epithelial cells are not restrained to form geometrical shapes that only consist of planar assemblies of outer surfaces, that the cells in fact can have a variety of curved (!) and convoluted shapes, even more complex than assumed here, and that depending on the epithelial tissue analysed cells can also be very malleable and accommodating, even at steady state, but especially dynamically.

Overall, the manuscript needs extensive re-writing, the first and second part are very disjointed, i.e. the voronoi modelling and biological examples are not well integrated with the energy-minimisation modelling and rather feel like to separate studies stuck together.

In addition, there are many grammatical errors (singular-plural disjunctions, illogical sentence structure etc.), the discussion is very repetitive and states the same argument several times in slightly different phrasing.

Scutoids are a geometrical solution to three-dimensional packing of epithelia

Point-by-point response to the reviewers' comments:

Reviewer #1, an expert in epithelial morphogenesis (Remarks to the Author):

This paper from the groups of Buceta & Escudero reveals that apico-basal topological changes are expected in epithelia with differential curvature. The authors neologise the incomplete cell-cell interfaces involved as 'scutoids'. The paper starts by demonstrating the 3D phenomenon with the help of Voronoi tessellation on epithelial tubes, proceeds to show that scutoids are seen in three real curved epithelia, and concludes with a more formal physical approach to predict when scutoids are likely to appear, based on cell-cell interface length minimisation.

The paper makes a significant contribution to a question that has not yet been explicitly answered in the field: what is the 3D structure of cells in curved epithelia? This needs to be understood in order to understand 3D morphogenesis.

Overall the paper is interesting and potentially of interest to a high profile journal such as Nat Comms, though there are significant issues.

Answer: We thank the reviewer for his/her positive comments and interest in our manuscript. Changes in response to his/her comments and criticisms are highlighted in orange colour in the new version of the manuscript.

Major

1. In convergence-extension movements, such as axis extension in chick, zebrafish and *Drosophila*, new cell-cell contacts are actively made as a result of planar-polarised Myosin-II activity. It is widely accepted in this field that cells are likely to make new contacts at particular depths, not as prisms. For example, their ref 55 explicitly measures new contacts being made in the single-layered germ-band either on the apical or basal side, reflecting a long-standing question in this area. Hisao Honda has also modelled non-prismatic (scutoid-based) intercalations (Dev Dyn 2008 DOI: 10.1002/dvdy.21609 Fig. 3), though not explicitly in relation to curvature. Incomplete 3D cell-cell interfaces, or scutoids, are therefore not new. However, that they are predicted specifically because of the curvature of epithelia is to our knowledge new and interesting. Greater care is needed in not overemphasizing the novelty of their scutoids in various places – in the hyperbole of the abstract and in some confusion over the cause of scutoids they report (whether they are driven actively or the passive response to curvature).

Answer: Following the advice of the reviewer, we have toned down our claims about the novelty of the apico-basal intercalations. We have now introduced,

referenced and discussed other cases where epithelial cells present incomplete 3D cell-cell interfaces.

2. The specific novelty they report is that epithelial curvature generates scutoids. However, the precise relationship between curvature and the likelihood of scutoids is incompletely presented. The property of curvature that makes scutoids likely is anisotropic curvature, not curvature itself – this is not made clear and their ‘theoretical’ and ‘biophysical’ models do not explore the relationship between different curvatures in orthogonal directions. The salivary gland is a nice example of an epithelial tube with curvature in one orientation only, but each of the other 3 tissue examples presented have curvature in both orientations. The zebrafish EVL wraps an essentially spherical embryo, where prisms would be predicted because curvature in orthogonal orientations would be equal. The *Drosophila* egg chamber and germ-band tissues have principal and secondary axes of curvature, with the latter non-zero. Their predictions of scutoids as a result of curvature needs to explicitly account for the combinations of orthogonal curvature. It is not sufficient to claim that ‘some’ scutoids will be found in curved epithelia in a paper in a high profile journal. The underlying relationship needs to be fully described. In some of the epithelia analysed (germ-band, EVL), active cell intercalation is occurring, which involves scutoids (e.g. ref 55). How do the authors propose to disentangle situations where scutoids due to curvature and active processes are involved?

Answer: We agree with the reviewer about these comments. A similar comment has been raised by the reviewer 3. We have added new data to the manuscript to address these points:

- We have analysed in depth stage 4 and stage 8 *Drosophila* Egg chambers to obtain information about the 3D packing in a tissue with curvatures along two orthogonal directions.
- We have quantified the percentage of scutoids in the 50% epiboly zebrafish embryo (Text, **Supplementary Fig. 4, Supplementary Table 2 and Methods**).
- We have developed a “Voronoi spheroidal model” that accounts for different combinations of orthogonal curvatures. This also includes the case of the sphere where there are not scutoids.
- As the reviewer suggests, we now present our results focusing on the surface ratio anisotropy.

We have also discussed the possible reasons that could explain the differences between the Voronoi model and the actual tissue in the case of the salivary gland (where the tubular model presents a higher number of apico-basal transitions) and the stage 4 egg chambers (where the Voronoi spheroidal model present a lower number of apico-basal transitions).

We now propose that the model can identify the fraction of scutoids that appear due to the anisotropic curvature. In the cases where the model presents a lower number of scutoids, it should be additional active morphogenetic events (migration, convergent extension, proliferation) inducing the appearance of apico-basal intercalations.

Moreover, new figure 6b accounts for the value of the surface ratio anisotropy as measured by the local changes of the cellular aspect ratio. In that context we discuss the appearance of scutoids in different geometries within a unified framework.

3. It was not quite clear why both the Voronoi and biophysical approaches were needed. I agree that some minimisation of interface surface area is probably a sensible way to think about predicting when a transition is likely to occur in a 'passive' epithelium (i.e. one without basal or planar polarised Myosin, for example), as has been employed for the 'biophysical' approach. Since Voronoi tessellation minimises the total lengths of junctions, this approach is also 'biophysical' in the sense in which the authors use it, and wouldn't it do as good a job as their 'biophysical' approach? I suspect it would give very similar predictions. A vertex-based model would be another contender, where the perimeter and interface length terms would provide (effective) mechanical control over the frequency of transitions in a stretched sheet (as in Fig. S1). For example, nearly 2 decades ago, Chen & Brodland (2000, J Biomech. Eng.) in Fig. 3 explore parameter space for transitions in a stretched sheet (here also with orthogonal convergence, but the principle is good), much like Fig. S1. Can the authors better justify why both models were necessary, their relative advantages, what is a theoretical model that isn't biophysical, and discuss what alternative approaches would likely bring, such as vertex-based/finite element approaches. The paper reads as though fig 4 was added separately, while it would be better if the unique aspects of Voronoi and biophysical models could be interwoven.

Answer: We have interwoven both theoretical models and presented our results in a more coherent way. Also, we now estimate the tensile energy in Voronoi models (tubes, spheroids) to better compare computational and actual tissues. We discuss in the text the logic and benefits (and also the limitations) of using the line-tension energetic model.

In addition, we now place our theoretical/computational framework in a broader context that explain how the surface ratio anisotropy induce the appearance of apico-basal intercalations in both Voronoi model and curved epithelia. The new discussion section includes the study that the reviewer mentioned that shows a direct correlation between changes in the cellular aspect ratio and cellular rearrangements in tissues.

4. In Fig 2 and associated Fig S1 the effect of cell number (10-400 cells) on "transitions" is surely nothing to do with cell density but due to edge effects? If done with periodic boundaries along the tube, or for longer tubes, the differences between the lines in Fig 1B would vanish? If this is correct, it doesn't inspire confidence in the formal aspects of the paper.

Answer: Following the referee comment, and in order to ensure that finite size effects do not introduce any artefact in our analysis, we have discarded the cells in the edges. Also, we have extended the length of the Voronoi tubes (fourfold) yet keeping the same cell density. The new analysis confirms, but also improve,

our previous results and show that the lower the cell density the smaller the amount of scutoids.

Minor

1. It was not at first obvious why the neologism ‘scutoid’ had been chosen, expecting it to describe the 3D shape of cells with apico-basal twists. It became clear that the term refers to the triangular cell-cell facets of cells whose topology changes between apical and basal, and these triangles are indeed reminiscent of the scutellum of a hemipteran. It would not be a bad idea to add a picture or dorsal diagram of a hemipteran (like [https://en.wikipedia.org/wiki/Scutellum_\(insect_anatomy\)#/media/File:Heteroptera_morphology-d.svg](https://en.wikipedia.org/wiki/Scutellum_(insect_anatomy)#/media/File:Heteroptera_morphology-d.svg)) to show this explicitly in the paper (by Fig. 1E?).

Answer: We thank the referee for this suggestion. We have included in the new version of Fig. 1 the picture of the thorax of a *Cetoniidae* (**Fig.1 e**) showing the resemblance of its scutum and scutellum with the scutoids.

2. In the abstract, the following words or phrases are either too strong or inappropriate: “general principle” too strong, “scholarly publications” avoid, “necessarily undergo” can undergo, “compelled to adopt” adopt. The second half of the final sentence is misplaced hyperbole.

Answer: We have included these changes in the abstract and hopefully improve the overall quality of the language and our narrative in the new version of the manuscript.

3. The quality of prediction in Fig. 3E is poor (why no stats?). Would something like edge length/centroid-centroid length have discriminated better? Is this figure superseded by Fig. 4F?

Answer: We have made several improvements to the manuscript regarding this point. First, we have now more data from “transitions” and “no transitions” in the case of the salivary gland and the other analysed tissues. In each type of sample, we have performed a statistical test (Kolmogorov-Smirnov) to analyse the level of dissimilarity of the “edge angles” and “edge length” distributions (**Table S4**).

In addition, we have performed a similar analysis using the Voronoi tubular models and the Voronoi spheroidal model to compare the results with actual samples.

Our new data support our argument that geometrical constraints drive the appearance of scutoids. While closely related, this argument is different to that shown in **Fig. 4f** (now **Fig. 4d, e**) since in that case we provide a plausible cause to that phenomena using energetic considerations.

4. The point of Fig 2D&E and the text that go with it are not clear, as they are not subsequently used – a formal relationship must exist between angles, interface lengths and likelihood of transition that would be more useful than simulated trends? The accompanying text could be more formal and precise.

Answer: We have changed these panels and now we show the data in the same format that the real samples (**Fig. 3e** and **Supplementary Fig. 2**). We have tried to find a direct formal relationship between angles, interface lengths and likelihood of transition relationship, but we have not been able to uncover it. For a given surface ratio anisotropy, it depends on the angle and the length of the interface in each four-cells motif. This is also related to the change of the aspect ratio as shown in the new Supplementary Fig 3.

5. P.8, line 15: this is not tracking in time, as would be expected from the word, so I would 'track in depth'. P.22, line 2: 'cell tacking' is 3D segmentation.

Answer: We have changed these lines in the text.

6. "Some", used in various places should be replaced with a quantity.

Answer: We have now included the quantities in the text.

7. That scutoids can be concave: agreed, but differential pressure between neighbouring cells, as used in the force inference methods by Brodland et al, also generates concave interfaces, and these are quite common.

Answer: We explain the concavity of scutoids as part of its geometrical description. Following the referee advice, we have now placed the geometrical description of scutoids in a broader context and we have included that reference (pag 13, line 19).

8. Fig S4C. A transverse view of the U-shaped furrow would help make sense of the panels shown, and to visualise the apical and basal curvatures.

Answer: We have modified **Supplementary Fig. 4b** by including a transverse view of the furrow region and adding the apical and basal labels to that panel.

9. Fig S4E. Not clear which is the surface orientation with greatest curvature, so not clear whether highlighted transitions are in line with predictions of differential curvature.

Answer: In the new version we have indicated explicitly the surface orientation in the different tissues analysed to make clear which is the surface with the largest curvature.

Reviewer #2, an expert in biological patterning and self organisation (Remarks to the Author):

The paper proposes a novel method of generating shapes of epithelial cells, based on Voronoi tessellations, that leads to more diverse and realistically looking forms of cells emerging in 3D packing of curved epithelia, compared with

commonly used prisms or frusta. The attention is concentrated on the novel feature of spacial intercalation that involves the appearance of a vertex along the apico-basal axis of the cells.

The paper is original and important and deserves publication but the presentation needs to be improved.

Answer: We thank the reviewer for his/her positive comments and interest in our manuscript. Changes in response to his/her comments and criticisms are highlighted in dark blue colour in the new version of the manuscript.

The arrangement of the paper makes reading and understanding difficult.

Answer: We have performed a thorough reorganization of the narrative of the manuscript following the advice of the different reviewers.

The meaning of apico-basal transition is not well defined in the text. It appears to be used interchangeably with apico-basal cell intercalation and becomes better understood only in the Discussion where it is characterised as a spacial T1 transition.

Answer: We now provide a better and earlier introduction to the interchangeable concepts of “transitions” and “intercalations” using the T1 analogy (pag 5, line 28).

The "computed nine conditions" are not listed explicitly.

Answer: The nine conditions that have been used refers to the nine curvature ratios listed. We have clarified this part on the text (pag 6, line 27).

The authors present a long sequence of Results, with a plethora of rather obscure terminology, and statements which the reader is supposed to blindly believe, while the essential mechanical theory is hidden in the Materials and Methods part stuck in the end, and, unfortunately, is far from being adequate. Some details, like technicalities of the construction of Voronoi diagrams, preparing samples, and imaging, can be indeed hidden in this Appendix, but the energy considerations, appearing in the very end, are crucial. Voronoi diagrams are just a formal device, and only mechanical and biological evidence can justify the stated preponderance of the novel cellular shapes.

Answer: We have included an explanation and justification of the use of the line-tension model. This is supported by the **Fig. 4** and by the **Methods** sections where we provide technical details. We have tried to avoid as much as possible the technicalities in the main text to ease the reading for a general audience. Still, following the advice of the referee, we better justify our findings in terms of mechanical and biological evidence. We believe that the new narrative and figures of the manuscript makes the understanding easier.

The energy computations are the weakest point of the paper, and the expressions in this part lack rational justification. They are based on the notion of line tension, which is a marginal feature that is largely considered to be irrelevant or at least not fundamental in continuum mechanics. This analysis should involve surface and bulk energies, analogous to line and area energies in the theory of 2D tessellations. These energies are not easy to quantify, as they depend on biological factors, such as cell adhesion, actin stress, and osmotic pressure. This analysis may remain outside the scope of this paper, as comprehensive theory might be outside the domain of expertise of this group or require long time to develop. I will support publication if the energy analysis is omitted, provided the paper and Supplementary Material contain sufficient information for the analysis by interested researchers.

Answer: We acknowledge that a more realistic mechanical description of the energetic consideration would include not just line tension but surface stress, apical constriction, and pressure terms among others. Yet, as the referee points out, those are not easy to quantify. We now highlight better the limitations of our approach. Yet, regardless of the undeniable contribution of those terms to the cell energy, we want to stress that as in the case of T1 transitions, tensile contributions are all-important and provide the minimal energetic framework to understand the observed phenomenology. We better justify these arguments in the revised version of the manuscript.

The paper has a number of linguistic problems:

- The term "scutoids" is obscure and not suggestive of the generated forms that do not look as scales. Perhaps, "prismoids" is a better fit.

Answer: We have chosen a new word since the "scutoids" geometrical shape it is really different of prisms and prismatoids, due to the additional vertex. The term prismoid is has been already defined to refer to "a prismatoid that has planar sides, and the same number of vertices in both of its parallel planes". In **Fig. 1e** we now show the picture of the thorax of a *Cetoniidae* that emphasizes the resemblance of the scutum and scutellum with the geometrical shape we, accordingly, have termed scutoids.

- The terms "apical" and "basal" are interchanged relative to the common use; it would be proper to call the outer surface apical, and in Figs.1,3 the apical side is below and the basal is above.

Answer: In the salivary gland (**Fig. 2**), the inner surface is formed by the apical part of the epithelial cells. For this reason, in the **Fig. 1** and **3**, we have used this terminology for the Voronoi tubular model. We have preferred to choose this terminology for the sake of consistency in the comparisons between the model and the actual tissue. In the egg chambers, we have the same case. The follicular cells also present the apical part towards the interior of the spheroid (**Fig. 5**). Since each tissue organize in a diverse way, we have indicated in a clear way

the location of apical and basal surfaces in the different panels of **Supplementary Figure 4**.

- p.7: obscure expressions: "enrichment of values", "apparition of transitions" (appearance? emergence?)

Answer: We have changed these sentences and hopefully improve the quality of the language in the new version.

- p.14: "contacts between the cells are not convex" - The authors apparently mean that the sign of the mean curvature varies along the contact surface.

Answer: We have changed this sentence to explain that the surfaces of scutoids can be concave or convex and that this allow the 3D packing along the apico-basal axis.

- p.17: "The metric in each one of the surfaces ... is just the distance of the shortest geodesic joining two points" - this might be the way the authors define distances but it is not the standard definition of the metric, which is a tensor.

Answer: In the field of computational geometry the distance between points is usually considered as the length of the shortest geodesic. Therefore, we have considered that this is the best way to define the metric in this paper.

- p.5: a funny typo: To spam a wide range of basal expansions.

Answer: We have changed this sentence.

Reviewer #3, an expert in epithelial morphogenesis (Remarks to the Author):

This manuscript predicts the presence of a 'novel' 3D epithelial cell shape that the authors term 'scutoid'. Voronoi modelling of cell packing in tubular structures at both apical and basal surfaces predicts transition of cell contacts along the apical-basal axis, termed intercalation, leading to these more complex shapes. The authors then go on to analyse tubular tissues, segment cell shapes and show that indeed scutoids can be found in many tissues. In the second part of the manuscript they use a model of line tension and energy minimisation to predict that energetically a packing involving transitions or scutoids is also favourable.

A few general comments first: the most interesting aspect of the paper in my mind is that the authors show that in tubular tissues of a certain radius ratio, space filling packing of epithelial cells requires the appearance of complex shapes such as scutoids, but overall this paper does not clearly state the case as to why this should also be true for other systems of epithelial bending that are not cylindrical

tubes. Extensive rewriting and restructuring could make the manuscript more accessible and convincing.

Answer: We thank the reviewer for these positive comments. We have incorporated changes related to them into the manuscript text in green colour. We have changed thoroughly the structure and the narrative of the manuscript to make it more accessible. We also have included new data regarding systems with two curvatures and discussed the appearance of apico-basal intercalations in them (see below).

Also, part of what the authors suggest has been observed qualitatively previously, i.e. intercalation along the apical-basal axis itself in epithelia is not novel and in fact has been described as early as 1991 by Condic, Fristrom and Fristrom (Apical cell shape changes during *Drosophila* imaginal leg disc elongation: a novel morphogenetic mechanism. *Development*, 111(1), 23–33.) in case of *Drosophila* leg discs.

Answer: We thank the reviewer for this reference. We have tone down our claims of novelty and included this reference together with others that described this type of phenomena before.

What the authors add here is the quantitative aspect. The authors make a good case of examining the salivary gland example, but what I find somewhat confusing is why scutoids are also found in several of the other examples such as follicle cells in stage 4 egg chambers and the EVL during zebrafish epiboly. The geometric constraints of these tissues are very different to the theoretical voronoi cylinder as well as the salivary gland model described first, and this is not properly discussed. Especially in the case of epiboly, the overall shape is near spherical, i.e. similar amounts of curvature to the epithelium in all directions, whereas a cylinder by definition is curved only in one of two orthogonal directions. So would voronoi modelling in such case also predict scutoids?

Answer: We agree with the reviewer about these comments. A similar comment has been raised by the reviewer 1. We have added new data to the manuscript to address these points:

- We have analysed in depth stage 4 and stage 8 *Drosophila* Egg chambers to obtain information about the 3D packing in a tissue with curvatures along two orthogonal directions.
- We have quantified the percentage of scutoids in the 50% epiboly zebrafish embryo (Text, **Supplementary Fig. 4**, **Supplementary Table 2** and **Methods**).
- We have developed a “Voronoi spheroidal model” that account for different combinations of orthogonal curvatures. This also includes the case of the sphere where there are not scutoids.

Scaling of configurations from apical to basal in a spherical case should suffice to lead to a space-filling epithelial packing. Isn't it more likely that the intercalations along the apical-basal length observed in the case of epiboly are

consequences of the dynamic behaviour of the tissue, i.e. snap shots of events that occur during more complex transitions of dynamic epithelial tissues? The same could explain these intercalations found in the follicular cells surrounding the germ line, as these begin a highly dynamic behaviour of rotation around the germline at stage 4 that is shown here (Cetera, M., Juan, G. R. R.-S., Oakes, P. W., Lewellyn, L., Fairchild, M. J., Tanentzapf, G., et al. (1AD). Epithelial rotation promotes the global alignment of contractile actin bundles during *Drosophila* egg chamber elongation. *Nature Communications*, 5, 1–12. <http://doi.org/10.1038/ncomms6511>).

Answer: We strongly agree with the reviewer about the importance of this possibility. The salivary gland is a tissue that does not present cell division or rearrangement at the stage that we are analysing the samples (third instar larva). On the contrary, as mentioned by the reviewer, the follicular epithelium at the stage 4 egg chamber is a very dynamic tissue. We now think that the appearance of scutoids can be also due to the dynamics rearrangements of the epithelial tissue and we have introduced this concept in the discussion. Interestingly, some of our new data support this possibility, since the stage 4 egg chambers present a higher number of apico-basal transitions than the corresponding the Voronoi spheroidal model. These data have been added to the results section and deeply discussed in the new version of the manuscript. We are really thankful to the reviewers for these comments since these findings open new promising future directions.

We now propose that the model can identify the proportion of scutoids that appear due to the anisotropic curvature. In the cases where the model presents a lower number of scutoids, it should be additional active morphogenetic events (migration, convergent extension, proliferation) inducing the appearance of apico-basal intercalations.

Moreover, new figure 6b accounts for the value of the surface ratio anisotropy as measured by the local changes of the cellular aspect ratio. In that context we discuss the appearance of scutoids in different geometries within a unified framework.

Another consideration that should at least be discussed is that as a general point, I would argue that because epithelial cells are not restrained to form geometrical shapes that only consist of planar assemblies of outer surfaces, that the cells in fact can have a variety of curved (!) and convoluted shapes, even more complex than assumed here, and that depending on the epithelial tissue analysed cells can also be very malleable and accommodating, even at steady state, but especially dynamically.

Answer: We have included this point in the Discussion section (Pag 14, line 22). We agree with the reviewer that columnar epithelial cells (for example in the *Drosophila* imaginal discs) can for sure present shapes even more complex than scutoids. Indeed, we show in **Supplementary Fig. 2a** how the increase of the surface ratio correlates with the appearance of multiple apico-basal transitions in

each cell. These cells will present multiple contacts at different levels with the neighbouring cells. In these cases, the cells squish between others to establish the 3D packing the tissue.

In addition, regarding to this point and to the previous one, we have discussed that the morphogenetic dynamics of the tissue (proliferation, migration, convergent extension) can induce apico-basal intercalations even without curvature.

Overall, the manuscript needs extensive re-writing, the first and second part are very disjointed, i.e. the voronoi modelling and biological examples are not well integrated with the energy-minimisation modelling and rather feel like to separate studies stuck together.

Answer: We have changed the structure of the manuscript integrating the biological data and the theoretical/computational framework. This includes the new Voronoi spheroidal model and the egg chamber data. Also, we have performed an extensive re-writing to present our findings in a more coherent way.

In addition, there are many grammatical errors (singular-plural disjunctions, illogical sentence structure etc.), the discussion is very repetitive and states the same argument several times in slightly different phrasing.

Answer: The manuscript has been corrected by a native English speaker to eliminate these grammatical errors and we have done our best to improve our narrative and avoid repetitions.

REVIEWERS' COMMENTS:

Reviewer #1 (Remarks to the Author):

The authors have taken on board all suggestions and have made a thorough job of improving the manuscript. In particular, a thorough exploration of the impact of differential curvature on scutoid expectation, and the mixing of the Voronoi and biophysical approaches through the paper, that were previously separate, are now well done.

The paper now reads as a nicely nuanced treatment of scutoids in 3d epithelia. It doesn't matter in the least that, in the biological examples they've analysed, the number of T1s in depth don't accurately match their baseline predictions based on curvature alone. All tissues will be active in some way, but the authors provide a default expectation of what active processes will be pulling real tissues away from.

Short of a few minor wordy quibbles listed below, publication is recommended.

- first line of abstract: fly and fish embryos start out strongly curved, though chick/mouse are more flat - suggest something more general (and correct) saying tissue bending contributes to shape organs into complex 3D structures
- p9, l.22-25: this sentence is a little obscure, could be rephrased to be more biologist-friendly.
- not clear from only reading the main text what these trajectories are, would help to define them in the main text. What are you measuring exactly? Would also help understand the above point.
- p.12 line 12: could add "bias due to differential curvature in the orientation..." ?

Reviewer #3 (Remarks to the Author):

This revised version of the manuscript is much improved, the different aspects of the study, Voronoi modelling, biological examples and energy minimisation approaches, are much better integrated and together make a convincing case for the arguments presented.

The authors have taken the various reviewers' recommendations on board and changed the manuscript accordingly.

I have a few remaining comments:

I still find that the authors slightly overstate their case, especially in the abstract:

"Hence, we conclude that scutoids are nature's solution to achieve epithelial bending. Our findings pave the way to understand the three-dimensional organization of epithelial organs."

I would suggest to rather state that scutoids are 'one of nature's solutions to achieve epithelial bending', as the authors themselves discuss that depending on the geometry, scutoids are required to accommodate bending or not, i.e. a sphere does not require scutoids, and real bent epithelia will be a combination of scutoids and frustra/wedges, whereas the sentence above implies that you only need scutoids.

page 3 line 24:

PCP is not a morphogenetic process as the authors state here.

throughout the manuscript, it should be made clear that the salivary glands analysed are larval salivary glands, especially in contrast to the embryonic epithelia analysed. Larval salivary glands are at steady-state and days post-morphogenesis. Please add 'larval' as a descriptor.

page 4 line 29:

same overstatement as in the abstract

page 5 line 19-21:

this sentence in its current state does not make sense:

"On the apical surface, neighbouring seeds along the longitudinal direction of the tube did not change the distance on the basal surface (Fig. 1c, green and red cells). "

Figure 4,d and e:

why is the 'no transitions' case for the in vivo salivary gland so different from the Voronoi model in terms of angle and spread of angle? (also true later for the spheroidal voronoi)

end of page 10, beginning of page 11, stage 4 to stage 8 comparison of egg chambers:

Isn't this a rather surprising result? Wouldn't you expect that going from a sphere (where for a perfect sphere you would not expect any transitions/scutoids as the Voronoi should scale perfectly between apical and basal surfaces) to an elongated spheroid that you increase the number of transitions/voronoi as the curvature along the main axes become more anisotropic? This is discussed in the discussion, but shouldn't the discrepancy between result and prediction be spelt out more clearly at this stage?

page 11, line 15:

Instead of 'The aforementioned experimental evidence suggests that scutoids are a general feature of curved epithelia.'

rather say

'...of many' or 'of certain curved epithelia'.

You discuss yourself that not all geometries require scutoids.

page 15, line 13:

please state more clearly what you mean with the 'actomyosin ring' in the brackets, does this refer to apical-junctional actomyosin? if so then state this, please.

There are still many many typos and grammatical inconsistencies that should be rectified, I only list a small fraction below:

page 4 line 14:

should be 'has' instead of 'have'

page 4 line 21:

should be 'tissues' not 'tissue'

page 10, line 28/29:

should be 'during the egg chamber maturation process' rather than 'during egg the chamber maturation process'

REVIEWERS' COMMENTS:

Reviewer #1 (Remarks to the Author):

The authors have taken on board all suggestions and have made a thorough job of improving the manuscript. In particular, a thorough exploration of the impact of differential curvature on scutoid expectation, and the mixing of the Voronoi and biophysical approaches through the paper, that were previously separate, are now well done.

The paper now reads as a nicely nuanced treatment of scutoids in 3d epithelia. It doesn't matter in the least that, in the biological examples they've analysed, the number of T1s in depth don't accurately match their baseline predictions based on curvature alone. All tissues will be active in some way, but the authors provide a default expectation of what active processes will be pulling real tissues away from.

Answer: We thank again the reviewer for his/her positive comments.

Short of a few minor wordy quibbles listed below, publication is recommended.

- first line of abstract: fly and fish embryos start out strongly curved, though chick/mouse are more flat - suggest something more general (and correct) saying tissue bending contributes to shape organs into complex 3D structures

Answer: We have changed the abstract following the reviewer's suggestions.

- p9, l.22-25: this sentence is a little obscure, could be rephrased to be more biologist-friendly.

- not clear from only reading the main text what these trajectories are, would help to define them in the main text. What are you measuring exactly? Would also help understand the above point.

- p.12 line 12: could add "bias due to differential curvature in the orientation..." ?

Answer: We have changed the abstract accordingly to the reviewer suggestions.

Reviewer #3 (Remarks to the Author):

This revised version of the manuscript is much improved, the different aspects of the study, Voronoi modelling, biological examples and energy minimisation approaches, are much better integrated and together make a convincing case for the arguments presented.

The authors have taken the various reviewers' recommendations on board and changed the manuscript accordingly.

Answer: We thank again the reviewer for his/her positive comments.

I have a few remaining comments:

I still find that the authors slightly overstate their case, especially in the abstract:

“Hence, we conclude that scutoids are nature’s solution to achieve epithelial bending. Our findings pave the way to understand the three-dimensional organization of epithelial organs.”

I would suggest to rather state that scutoids are ‘one of nature’s solutions to achieve epithelial bending’, as the authors themselves discuss that depending on the geometry, scutoids are required to accommodate bending or not, i.e. a sphere does not require scutoids, and real bent epithelia will be a combination of scutoids and frustra/wedges, whereas the sentence above implies that you only need scutoids.

page 3 line 24:

PCP is not a morphogenetic process as the authors state here.

throughout the manuscript, it should be made clear that the salivary glands analysed are larval salivary glands, especially in contrast to the embryonic epithelia analysed. Larval salivary glands are at steady-state and days post-morphogenesis. Please add ‘larval’ as a descriptor.

page 4 line 29:

same overstatement as in the abstract

Answer: We have made all these changes following the reviewer’s suggestions.

page 5 line 19-21:

this sentence in its current state does not make sense:

“On the apical surface, neighbouring seeds along the longitudinal direction of the tube did not change the distance on the basal surface (Fig. 1c, green and red cells).”

Answer: We have rewritten the sentence to make the text clearer.

Figure 4,d and e:

why is the ‘no transitions’ case for the in vivo salivary gland so different from the Voronoi model in terms of angle and spread of angle? (also true later for the spheroidal voronoi)

Answer: As mentioned in the text (pag. 13, line 32), one of the limitations of our computational model is that we neglect some biophysical characteristics of the cells, such as their stiffness or surface stress. We hypothesize that those differences are due to those and also to the unavoidable role of noise: e.g. thermal noise, or noise in gene expression that implies variability in the concentration of the molecular effectors that are the ultimate responsible of promoting a neighbour exchange. That variability is clear in Fig S3: compare the spread of the density functions.

end of page 10, beginning of page 11, stage 4 to stage 8 comparison of egg chambers: Isn’t this a rather surprising result? Wouldn’t you expect that going from a sphere (where for a perfect sphere you would not expect any transitions/scutoids as the Voronoi should scale perfectly between apical and basal surfaces) to an elongated spheroid that you increase the number of transitions/voronoi as the curvature along the main axes become more anisotropic? This is discussed in the discussion, but

shouldn't the discrepancy between result and prediction be spelt out more clearly at this stage?

Answer: We understand that a comment on these lines could be made in the Results section. However, we decided that it will be more clear and easy to read if we put together the considerations related to the differences between dynamic tissues and Voronoi models in the Discussion section.

page 11, line 15:

Instead of 'The aforementioned experimental evidence suggests that scutoids are a general feature of curved epithelia.'

rather say

'...of many' or 'of certain curved epithelia'.

You discuss yourself that not all geometries require scutoids.

page 15, line 13:

please state more clearly what you mean with the 'actomyosin ring' in the brackets, does this refer to apical-junctional actomyosin? if so then state this, please.

Answer: We have made all these changes accordingly to the reviewer suggestions.

There are still many many typos and grammatical inconsistencies that should be rectified, I only list a small fraction below:

page 4 line 14:

should be 'has' instead of 'have'

page 4 line 21:

should be 'tissues' not 'tissue'

page 10, line 28/29:

should be 'during the egg chamber maturation process' rather than 'during egg the chamber maturation process'

Answer: We have proofread the text again to correct typos and grammatical mistakes.